# VLM's Eye Examination: Instruct and Inspect Visual Competency of Vision Language Models

**Nam Hyeon-Woo**                                                      *hyeonw.nam@postech.ac.kr*
*Department of Electrical Engineering*
*POSTECH*

**Moon Ye-Bin**                                                          *ybmoon@postech.ac.kr*
*Department of Electrical Engineering*
*POSTECH*

**Wonseok Choi**                                                         *wonseok.c@postech.ac.kr*
*Grad. School of AI*
*POSTECH*

**Lee Hyun**                                                          *hyun103.lee@samsung.com*
*Department of Electrical Engineering, POSTECH*
*Samsung AI Center, Samsung Electronics*

**Tae-Hyun Oh**                                                      *thoh.kaist.ac.kr@gmail.com*
*School of Computing, KAIST*
*Department of Electrical Engineering & Grad. School of AI, POSTECH*

**Reviewed on OpenReview:** *https://openreview.net/forum?id=CgWkVb2lHB*

## Abstract

Vision language models (VLMs) have shown promising reasoning capabilities across various benchmarks; however, our understanding of their visual perception remains limited. In this work, we propose an eye examination process to investigate how a VLM perceives images, focusing on key aspects of visual recognition, ranging from basic color and shape to semantic understanding. We introduce a dataset, LENS, to guide VLMs to follow the examination and check its readiness. Once the model is ready, we conduct the examination. We quantify and visualize VLMs' sensitivities to color and shape, and semantic matching. Our findings reveal that VLMs have varying sensitivity to different colors while consistently showing insensitivity to green across different VLMs. Also, we found different shape sensitivity and semantic recognition depending on LLM's capacity despite using the same fixed visual encoder. Our analyses and findings have the potential to inspire the design of VLMs and the pre-processing of visual input to VLMs for improving application performance.

## 1 Introduction

Vision language models (VLMs) (Liu et al., 2023b; Dai et al., 2023; OpenAI, 2023; Chen et al., 2023) are composed of a visual encoder to process visual information with a large language model (LLM) for comprehension, akin to how the human visual system operates with the eyes and brain. While VLMs have shown promising performance on various tasks (Marino et al., 2019; Mishra et al., 2019; Sidorov et al., 2020; Krishna et al., 2016), our understanding of how these models perceive visual information remains limited. Prior works (Choe et al., 2022; Zhou et al., 2016; Akata et al., 2023; Prystawski et al., 2023; Zhu & Li, 2023; Allen-Zhu & Li, 2023) have tried to understand the behavior or decision of neural networks, which would help to achieve responsible AI including explainability and safety. As the need for model understanding is

Figure 1: **Eye examination.** The process of eye examination contains three steps: instruction, readiness check, and examination. If a VLM has acquainted instructions and is ready, the model conducts examinations of color, shape, and semantic comparisons to assess its visual competency.

becoming increasingly important alongside significant advances in VLMs, we raise a fundamental question: *How do VLMs see and recognize?*

VLMs can understand questions and answer them in human-understandable language; therefore, we propose an eye examination process for VLMs by directly asking them questions to assess their visual recognition capabilities. However, directly asking VLMs unfamiliar questions without providing prior information about the examination does not yield meaningful results. Inspired by human vision testing, where participants receive instructions about how to conduct the test before the examination, we design a similar protocol: (1) Instruction – let a VLM know how the eye examination will proceed, (2) Readiness check – verify if the VLM is ready, and then (3) Examination – conduct the eye examination with questionnaires (refer to Fig. 1). For (1) instruction and (2) readiness check, we introduce a synthetic dataset named LENS (Learning ElemeNt for visual Sensory), categorized into basic visual elements such as color, shape, and semantics.

After the model passes the readiness check, we assess its recognition ability by comparing the reference and target objects in the step (3) examination. For example, in the color test, we ask if the VLM can distinguish subtle color differences such as **standard red** vs. **pinkish-red**. By conducting this comparative analysis, we can assess the sensitivity of VLMs to specific visual elements. For systematic analysis, we define the metrics of sensitivity: Sensitivity Area of Color (SAC) and Sensitivity Area of Shape (SAS). This reveals that the sensitivity levels are different for the reference colors and reference shapes. For semantic elements, we devise a patch-wise comparative analysis to reveal the semantic sensitivity of the VLM.

We perform the eye examination on LLaVA (Liu et al., 2023a) and InstrcutBLIP (Dai et al., 2023); we also employ advanced VLMs (Chen et al., 2024c; Wang et al., 2024; Marafioti et al., 2024) that do not require additional fine-tuning. The examination results reveal that the models have similar trends in visual competency, such as being less sensitive to the green color and having a similar tendency to shape. We include comparative analyses between human visual cognition and VLMs and investigate the effect according to the different scales of the LLM module. Our examination helps to understand the models' resolution in distinguishing similar colors and shapes. As a potential application, in chart image understanding, we can apply a simple preprocess to visual input, considering the model's sensitivity, so that we can improve the ability of VLMs to recognize chart images.

## 2 LENS Dataset for Instruction and Readiness Check

Understanding primitive competency at different levels is fundamental to vision perception as it is the basic building block of human cognition (Von Glasersfeld, 1989; Marr, 2010; Lowe, 2012). To understand the visual

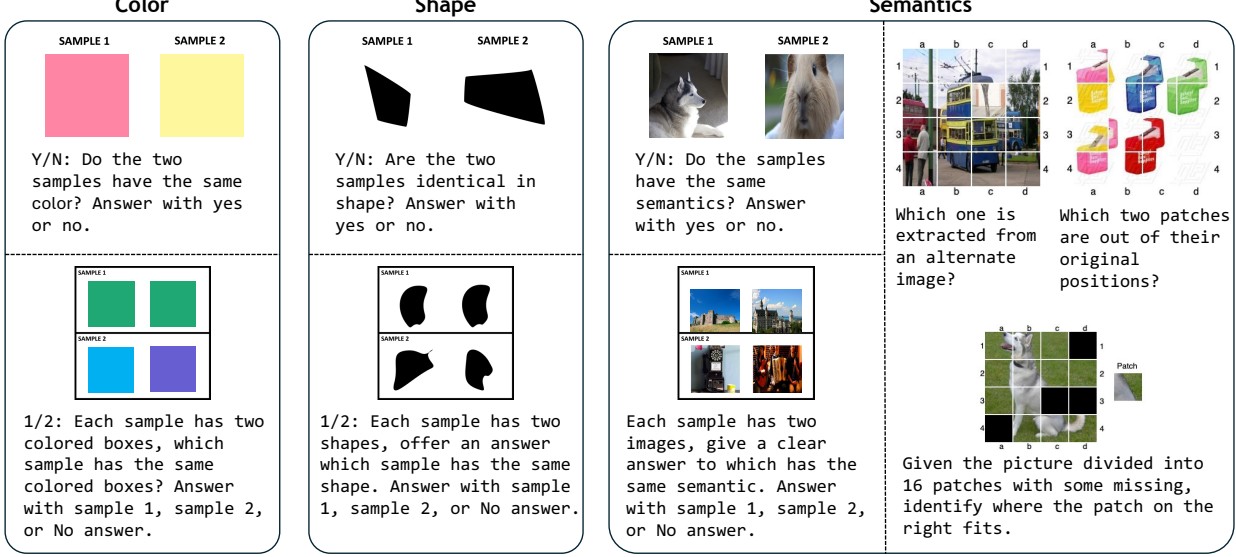

Figure 2: **LENS dataset.** We visualize the samples in LENS, which is designed to instruct VLM and check its readiness. LENS contains three categories: color, shape, and semantics. Note that questions for each sample are randomly sampled from a pre-defined question set, and the prompts for patches contain two options along with a no answer option. More details and data statistics can be found in the Appendix.

competency of VLMs, we propose an eye examination process involving three main steps: 1) instruction, 2) readiness check, and 3) examination. For steps 1 and 2, we introduce a dataset called LENS (Learning ElemeNt for visual Sensory), which has three primitive element categories: color, shape, and semantics. To give an instruction to a model about how to perform the examination, we finetune VLMs using LoRA (Hu et al., 2022) on the training set of LENS. Then, the test set of LENS is utilized to check the model's understanding of the instructions. Each sample of LENS data consists of an image, a question with answer options, and its ground truth answer (see Fig. 2). The questions are randomly sampled from a set associated with each category. If the model performs well on the LENS test set, we proceed to examine the model. The LENS format is also utilized in the examination stage. The examination step is described in Secs. 3.1-3.5.

## 2.1 Color LENS

The color element is designed to address specific queries categorized into either *yes/no* or *Sample 1 or Sample 2*, as shown in the first column of Fig. 2. The *yes/no* question type involves pairs of colors, prompting an assessment of whether the two colors are identical. The model should respond with yes or no, called *format*. The *Sample 1 or Sample 2* question type includes two sets, each containing two colors. The model should choose the correct sample or respond with no answer if both samples have different colors. The challenge lies in determining which sample pair accurately matches in terms of color. Note that we provide two separate color sets for training and test data, respectively.

We finetune LLaVA-v1.5 (Liu et al., 2023a) and InstructBLIP (Dai et al., 2023) with LoRA (Hu et al., 2022) on our color LENS training data. In Table 1a, the performance of both LLaVA and InstructBLIP becomes higher after finetuning and reaches acceptable levels on our color test set, regardless of their model size. This higher performance indicates that the models understand the instructions and are ready for the examination regarding color. The instructions and readiness check steps are performed in the same way for shape and semantic elements as for color. To measure visual competency on a particular element, we use a model separately finetuned on the training set for that element. We will examine visual perception in terms of color in Sec. 3.1.

Table 1: **Readiness check.** We evaluate LLaVA and InstructBLIP on the LENS test set after fine-tuning them on the LENS training set. The accuracy improvement after fine-tuning implies that LENS enhances the model's ability to compare two samples, indicating the model is ready to take the eye examination. Y/N stands for yes or no, 1/2 for sample 1 or sample 2, and S and P for the semantic and patch groups, ft for fine-tuning, respectively.

(a) **Color**

| Model | Yes or No | 1 or 2 |
|---|---|---|
| Random | 50.00 | 33.33 |
| LLaVA-7B | 88.38 | 41.20 |
| + Color (ft) | 89.79 | 98.59 |
| LLaVA-13B | 96.48 | 32.39 |
| + Color (ft) | 95.42 | 99.65 |
| InstructBLIP-3B | 89.79 | 47.53 |
| + Color (ft) | 100.0 | 72.53 |
| InstructBLIP-7B | 50.00 | 50.00 |
| + Color (ft) | 100.0 | 100.0 |

(b) **Shape**

| Model | Yes or No | 1 or 2 |
|---|---|---|
| Random | 50.00 | 33.33 |
| LLaVA-7B | 58.27 | 39.05 |
| + Shape (ft) | 78.15 | 99.88 |
| LLaVA-13B | 61.07 | 48.45 |
| + Shape (ft) | 80.60 | 99.70 |
| InstructBLIP-3B | 61.43 | 50.00 |
| + Shape (ft) | 100.0 | 99.88 |
| InstructBLIP-7B | 52.44 | 50.00 |
| + Shape (ft) | 100.0 | 99.94 |

(c) **Semantics**

| Model | S-Y/N | S-1/2 | P-Cross | P-Self | P-Mask |
|---|---|---|---|---|---|
| Random | 50.00 | 33.33 | 33.33 | 33.33 | 33.33 |
| LLaVA-7B | 67.10 | 22.69 | 36.73 | 32.47 | 46.67 |
| + Semantics (ft) | 81.70 | 95.00 | 41.47 | 47.93 | 50.00 |
| LLaVA-13B | 70.70 | 25.19 | 18.60 | 35.00 | 22.53 |
| + Semantics (ft) | 81.90 | 94.42 | 98.60 | 73.53 | 50.73 |
| InstructBLIP-3B | 69.30 | 50.00 | 41.07 | 28.40 | 33.27 |
| + Semantics (ft) | 84.30 | 83.65 | 41.20 | 48.07 | 50.07 |
| InstructBLIP-7B | 49.70 | 46.54 | 20.80 | 35.27 | 24.80 |
| + Semantics (ft) | 84.60 | 88.08 | 40.47 | 48.33 | 50.27 |

## 2.2 Shape LENS

The format for the shape element is the same as the color element but with color boxes replaced by shape images. We create these shape images using Bezier curves. The process begins with generating random points that are smoothly connected. The hyperparameters include the number of points, the radius around points, and the smoothness of the curve. We ensure that these hyperparameters do not overlap between the training and test sets. The second column of Fig. 2 shows the shape samples, with the readiness check results in Table 1b. The shape element examination is detailed in Sec. 3.3.

## 2.3 Semantic LENS

The semantic element has two groups, semantic and patch. The last column in Fig. 2 shows the samples for these groups, where the left is the semantic group and the right is the patch group. Semantic groups follow the same format as color or shape datasets but use images from ImageNet (Deng et al., 2009). The patch group has 3 types of images: self-swap, cross-swap, and masking, as shown in Fig. 2. In all cases, we divide the images sampled from ImageNet into $4 \times 4$ patches. The self-swap requires finding the positions of swapped patches, the cross-swap requires identifying a patch from a different image. The masking requires locating the correct position for one of the missing patches. All questions in the patch group offer three options, including the option of no answer. We check the readiness on the semantic element in Table 1c. The examination of the semantic element is detailed in Sec. 3.5.

# 3 Eye Examination for VLMs

During the examination, we evaluate the discrimination abilities of VLMs by asking whether two samples are identical in the context of multiple levels of vision primitives. In this paper, we conduct an eye examination on LLaVA-v1.5 (Liu et al., 2023a) and InstructBLIP (Dai et al., 2023). We also provide the examination results of recent advanced VLMs without additional fine-tuning, such as InterVL2.5 (Chen et al., 2024c), Qwen2-VL (Wang et al., 2024), and SmolVLM (Marafioti et al., 2024).

## 3.1 Examination: Color

Color is an important feature for distinguishing objects and is fundamental to many aspects of computer vision (Gevers et al., 2012). By distinguishing subtle color differences, we can perceive detailed information about an object, such as its curvature. We explore how VLMs perceive and process a subtle difference in color information and understand VLMs' color resolution.

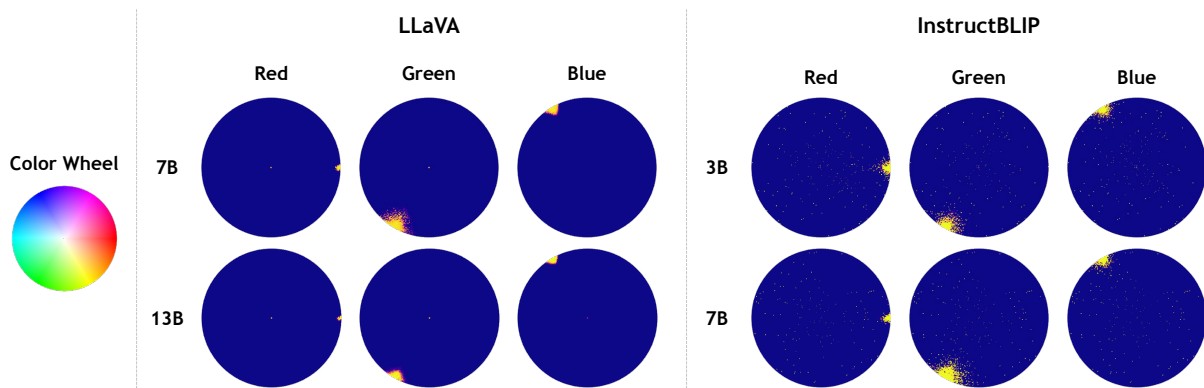

Figure 3: **Visualization Color sensitivity.** We measure the sensitivity of VLMs in differentiating between reference and target colors. Visualization of color sensitivity represented by $f(c_{\text{ref}}, c_{\text{target}})$, where $c_{\text{ref}}$ is one of red, green, and blue, and $c_{\text{target}}$ is selected from the color wheel. The result that green has the largest area of the wheel indicates distinguishing green is more challenging compared to red or blue.

We first look into how well VLMs can distinguish similar colors, utilizing the *yes or no* format. To measure color sensitivity, especially for the representative colors RGB, we proceed as follows:

1. Select a reference color $c_{\text{ref}}$ from red, green, and blue, and paint sample 1 with the selected $c_{\text{ref}}$.

2. Choose a target color $c_{\text{target}}$ from the HSV color wheel in Fig. 3, and use the chosen $c_{\text{target}}$ sample 2. The radius and angle on the wheel are divided equally into 100 and 500, respectively. This results in a total of 50k possible colors for $c_{\text{target}}$.

3. Ask VLMs whether the reference and target colors are the same. Then, record the token probability of "yes", denoted as $p(\text{"yes"}|c_{\text{ref}}, c_{\text{target}})$.

When the input has $c_{\text{ref}}$ and $c_{\text{target}}$, we extract token logits corresponding to "yes" and "no," and normalize the "yes" logit by these logits, denoted as the score function $f(c_{\text{ref}}, c_{\text{target}}) : \{c_{\text{ref}}, c_{\text{target}}\} \rightarrow [0, 1]$. In summary, color sensitivity is visualized by $p(\text{"yes"}|c_{\text{ref}}, c_{\text{target}}) = f(c_{\text{ref}}, c_{\text{target}})$. To quantify the visualized color sensitivity, we define it as follows.

**Definition 1. Sensitivity Area of Color (SAC).** Let $c_{\text{ref}}$ be the reference color, $c_{\text{target}}$ the target color, and $f(c_{\text{ref}}, c_{\text{target}}) : \{c_{\text{ref}}, c_{\text{target}}\} \rightarrow [0, 1]$ a score function that measures the similarity between colors. The function assigns a high value when the model recognizes that colors are similar.

$$\text{Sensitivity Area of Color} = \int f(c_{\text{ref}}, c_{\text{target}})dc_{\text{target}}. \tag{1}$$

However, since calculating the integral is not feasible, we use numerical integration as $\sum_{i=1}^{i=N} f(c_{\text{ref}}, c_{\text{target}})dA_i$ where $dA_i$ is the differential area. As shown in Fig. 3, we perform the calculation in polar coordinates, so $dA_i = rd\phi dr$ where $r$ and $\phi$ are the radius and angle, respectively. A low SAC value indicates that the model is capable of sensitively distinguishing the reference color. Conversely, a high SAC value stands for the model's insensitivity to the reference color.

**Result.** In Fig. 3, both LLaVA and InstructBLIP show high color sensitivity to red and low to green color. The visualization result in Fig. 3 shows that, although each reference color has a high probability in the neighborhood of each color, red is the most discriminating (sensitive), while green is the least discriminating (insensitive). The results are opposite to humans that are sensitive to green (Pasmanter & Munakomi, 2019; Robinson & Schmidt, 1984; Serway). Table 2 lists the SAC values that are consistent with the visualization. When comparing two models with the same size (7B), LLaVA is more sensitive to color distinctions. Advanced models without additional fine-tuning, specifically InterVL2.5, Qwen2-VL, and SmolVLM, generally align with the trend. The additional discussion can be found in the appendix.

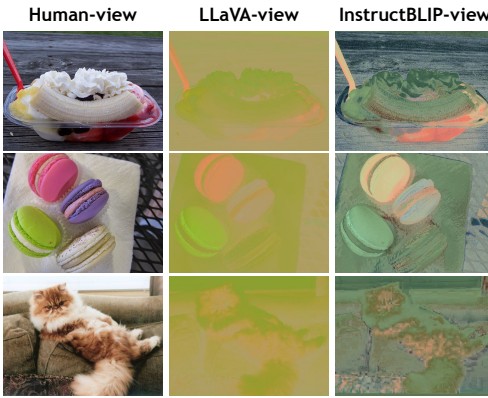

**Human-view** **LLaVA-view** **InstructBLIP-view**

Figure 4: **Color correction.** The first column is the original images that humans see. The second and third columns show the transformed images based on the color similarity patterns perceived by LLaVA and InstructBLIP, respectively. VLMs see the world greener.

Table 2: **Sensitivity Area of Color (SAC).** SAC quantifies the area of the wheel as defined in Definition 1. We denote fine-tuning as ft. Consistent with the visualizations in Fig. 3, the value for the green is the highest.

| Model | Size | Red | Green | Blue |
|---|---|---|---|---|
| LLaVA (ft) | 7B | 0.0064 | **0.0479** | 0.0156 |
| | 13B | 0.0036 | **0.0202** | 0.0105 |
| InstructBLIP (ft) | 3B | 0.0361 | **0.0533** | 0.0374 |
| | 7B | 0.0224 | **0.0793** | 0.0336 |
| InternVL2.5 | 2B | 0.1144 | **0.3953** | 0.1948 |
| | 4B | 0.0772 | **0.1827** | 0.1026 |
| | 8B | 0.0740 | **0.2023** | 0.1121 |
| SmolVLM | Instruct | 0.2487 | **0.8676** | 0.2652 |
| Qwen2 VL | 2B | 1.0727 | 0.9949 | **1.0868** |
| | 7B | 0.1108 | **0.3920** | 0.1592 |

Interestingly, while humans are sensitive to green (Pasmanter & Munakomi, 2019; Robinson & Schmidt, 1984; Serway), VLMs show the opposite result. Humans are more sensitive to medium wavelengths (often perceived as green) compared to long (red-sensitive) and short (blue-sensitive) wavelengths, resulting in heightened sensitivity to green. Inspired by this biological fact, we question the underlying factors leading to varying color sensitivities in VLMs. Specifically, we seek to understand the origins of these sensitivity variations in VLMs. Therefore, we investigate this further in the following section.

### 3.2 Why do VLMs have different color sensitivities?

What component does affect the color sensitivity of VLMs? We hypothesize that the capability of the visual encoder to perceive color significantly influences the decision-making process of LLMs, even more than LLMs themselves. To illustrate this, consider a thought experiment on the color sensitivity that converts the RGB value to the text and then ask for the probability from LLMs without a visual encoder. For instance, we can prompt like "Are (255, 0, 0) and (0, 255, 0) the same color? (Patel & Pavlick, 2022)" Since LLMs do not need to directly discriminate the colors themselves, we would expect the sensitivities to distinctive colors to be similar, leading us to question the role of the visual feature.

We measure cosine similarity between colors as follows: $sim(c_{\text{ref}}, c_{\text{target}}) = \frac{v(c_{\text{ref}}) \cdot v(c_{\text{target}})}{||v(c_{\text{ref}})||_2 ||v(c_{\text{target}})||_2}$, where $v(\cdot)$ is a visual encoder. We fixed the reference color as red, green, or blue and varied the target color within the 24-bit RGB color space. Considering the vast size of the color space, $256^3 \approx 16.8M$, we reduce the color space to $32^3 = 32,768$. We apply min-max normalization to ensure values within the range $[0, 1]$.

We evaluate CLIP ViT-L/14 336px (Radford et al., 2021) and ViT-g/14 (Fang et al., 2023), a visual encoder of LLaVA-v1.5 (Liu et al., 2023a) and InstructBLIP (Dai et al., 2023), respectively. When computing color similarity between hidden features for RGB and other colors, we notice that the pattern for green is opposite to that of red and blue. Details of the similarity pattern can be found in Fig. 11 in the Appendix. We use these patterns to convert the original image into an image as perceived by VLMs; we refer to this process as color correction. The transformation of the original RGB value is: $\tilde{\mathbf{I}}_{x,y} = (255, 0, 0) \cdot sim(c_{\text{Red}}, \mathbf{I}_{x,y}) + (0, 255, 0) \cdot sim(c_{\text{Green}}, \mathbf{I}_{x,y}) + (0, 0, 255) \cdot sim(c_{\text{Blue}}, \mathbf{I}_{x,y})$, where $\tilde{\mathbf{I}}_{x,y}$ and $\mathbf{I}_{x,y}$ stand for color-corrected and original image pixels.

**Result.** In Fig. 4, the results show how LLaVA and InstructBLIP perceive the world compared to the original images as perceived by humans. Surprisingly, the color-corrected images of both models appear

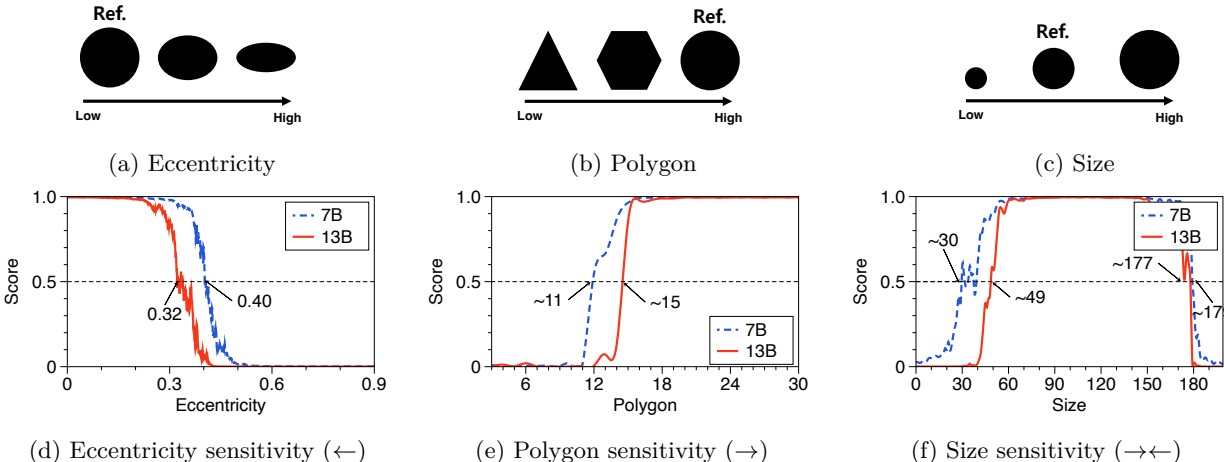

Figure 5: **Shape sensitivity.** We measure the sensitivity of VLMs between a circle and target shapes by varying **(a)** the eccentricity of a circle, **(b)** the number of vertices in a regular polygon, or **(c)** the size of a circle. The model is more sensitive if the score changes at **(d)** lower eccentricity, **(e)** a higher number of vertices, and **(f)** a narrower range of size. The results shows that a larger VLM is more sensitive than a smaller one.

greener than their original ones. The result implies that the models perceive the world as greener, enabling them to distinguish red and blue more than green. In summary, we examine the color cognitive ability. Our findings indicate that (1) both models are less sensitive to green, (2) this property is derived from the visual encoder, and (3) this differs from humans who are sensitive to green.

## 3.3 Examination: Shape

Fundamental features such as edges, corners, and blobs are extensively employed in feature descriptions Krig & Krig (2016); Mikolajczyk et al. (2003); Mukherjee et al. (2015). We investigate how VLMs process shape information. We examine the shape sensitivity similar to the color sensitivity in Sec. 3.1. Given the reference shape of a circle, the target shape is set by adjusting the circle's eccentricity, size, or the number of regular polygon vertices. As the eccentricity increases (see Fig. 5a), the number of vertices decreases (see Fig. 5b), or the size of the circle increases and decreases (see Fig. 5c), the target shape deviates from a reference circle, denoted as Ref. in each figure. We measure the probability of shape sensitivity as follows.

1. The reference shape $s_{\text{ref}}$ is a circle, which is sample 1.
2. Choose a target shape $s_{\text{target}}$ by varying eccentricity or the number of vertices. Eccentricity ranges from 0 to 0.9, divided into 900 segments. The number of vertices ranges from 3 to 30. The size has 200 levels, where 100 are smaller and the remaining 100 are larger than the reference shape. The chosen $s_{\text{target}}$ is depicted in sample 2.
3. Ask VLMs if the given shapes are the same, then record the token probabilities of "yes".

Based on the token probability of the shape sensitivity, we define the sensitivity area of a shape.

**Definition 2. Sensitivity Area of Shape (SAS).** Let $s_{\text{ref}}$ be the reference shape, and $s_{\text{target}}$ the target shape. We define a score function $f(s_{\text{ref}}, s_{\text{target}}) : \{s_{\text{ref}}, s_{\text{target}}\} \to [0, 1]$ that measures the similarity between shapes, assigning a high value when a model recognizes that shapes as similar.

$$\text{Sensitivity Area of Shape} = \int f(s_{\text{ref}}, s_{\text{target}}) ds_{\text{target}}. \tag{2}$$

For the feasible computation, we use numerical integration, where $ds_{\text{target}}$ is $1/1000$ for eccentricity, 1 for polygon, and 1 for size. Note that a low SAS value indicates that the model can sensitively distinguish between the reference circle and the target shape.

Table 3: **Sensitivity Area of Shape (SAS).** SAS quantifies the shape sensitivity as defined in Definition 2. We denote fine-tuning as ft. The result is consistent with the plot as shown in Fig. 5.

| Model | Model Size | Eccentricity | Polygon | Size |
|---|---|---|---|---|
| LLaVA (ft) | 7B | 0.4059 | 17.047 | 145.6 |
| | 13B | **0.3347** | **14.986** | **124.6** |
| InstructBLIP (ft) | 3B | 0.2084 | 10.73 | 80.4 |
| | 7B | **0.1662** | **9.74** | **69.6** |
| InterVL 2.5 | 2B | 0.6915 | 17.115 | 193.7 |
| | 4B | 0.4000 | **10.126** | 117.1 |
| | 8B | **0.3739** | 10.997 | **70.5** |
| Qwen2 VL | 2B | **0.2160** | 7.269 | 56.5 |
| | 7B | 0.2727 | **6.393** | **49.3** |
| SmolVLM | Base | 0.8544 | 25.785 | 191.7 |
| | Instruct | **0.4192** | **8.174** | **42.9** |

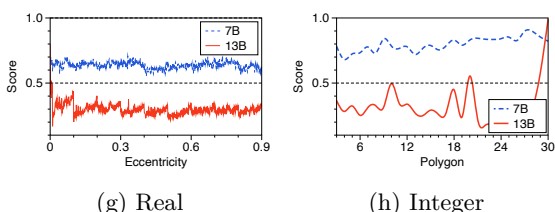

(g) Real    (h) Integer

Figure 6: **Shape sensitivity of LLMs.** The text prompts for two-number comparisons are given to LLMs. We extract and plot the probability score of the "yes" token for each prompt. **(a)** compares real numbers corresponding to eccentricity sensitivity. **(b)** compares integers for polygon sensitivity.

**Result.** We visualize the score function of LLaVA in Fig. 5d-5f, showing that a larger model is more sensitive than a smaller one. We mark the point when the first 0.5 score is achieved. In eccentricity sensitivity, the half-score points for LLaVA-7B and 13B are at 0.40 and 0.32, respectively. In polygon sensitivity, the half-score points are at about 11 and 15, respectively. In size sensitivity, the points are 30/179 and 49/177, respectively. InstructBLIP shows a similar trend to LLaVA, i.e., the larger the model, the more sensitive, which can be found Fig. 12 in the Appendix. As shown in Table 3, the SAS values also align with the graph results. Similarly, for InternVL2.5, the larger models generally are more sensitive compared to the smaller ones. Qwen2-VL also shows a similar overall tendency. For SmolVLM, the Instruct model shows higher sensitivity than the base model, which we attribute to its enhanced ability to follow the given instructions. This leads us to the question: why does shape sensitivity vary with model size?

### 3.4 Why do VLMs have different shape sensitivities?

What component affects the shape sensitivity? Since we use the same visual encoder regardless of model size, the difference likely stems from the capacity of the LLMs. To investigate this, we design an experiment to examine decision-making in LLMs by changing the text prompt and observing the token probability of shape sensitivity. The eccentricity prompt is "Is $\{r_1\}$ greater than or equal to $\{r_2\}$?," where $r_1 = 0$ and $r_2$ as a real number between 0 and 0.9. The polygon prompt is "Is $\{n_1\}$ greater than or equal to $\{n_2\}$?," where $n_2 = 3$ and $n_1$ as an integer between 3 and 30. We append "Answer with yes or no." to each text prompt and record the score of "yes" and "no" tokens.

**Result.** The scores for each text prompt are plotted in Fig. 6. We observe similar trends between Fig. 5g-5h and Fig. 5d-5e. The large LLM drops below 0.5 faster than the small LLM in Fig. 5d and Fig. 5g, and vice versa in Fig. 5e and Fig. 5h. We also notice that the smaller LLM is less accurate at comparing two numbers than the larger LLM. For example, the correct score should be below 0.5 in the first text prompt because 0 is less than other numbers in the range [0, 0.9], but 7B shows a higher score than 0.5, and it is also applied to the second text prompt. In summary, we investigate the shape recognition ability of VLMs as follows: (1) larger VLMs are more sensitive to shape, (2) the decision-making of LLM influences the shape sensitivity, and (3) the model size of LLMs influences the numerical comparison ability, which is connected to the shape sensitivity.

### 3.5 Examination: Semantics

Semantics represents a foundational component of vision recognition. Humans can classify objects into their semantic classes, regardless of variations in color or shape. This ability extends to perceiving partially obscured objects; we can infer the form of the hidden parts if we recognize the category. In this regard,

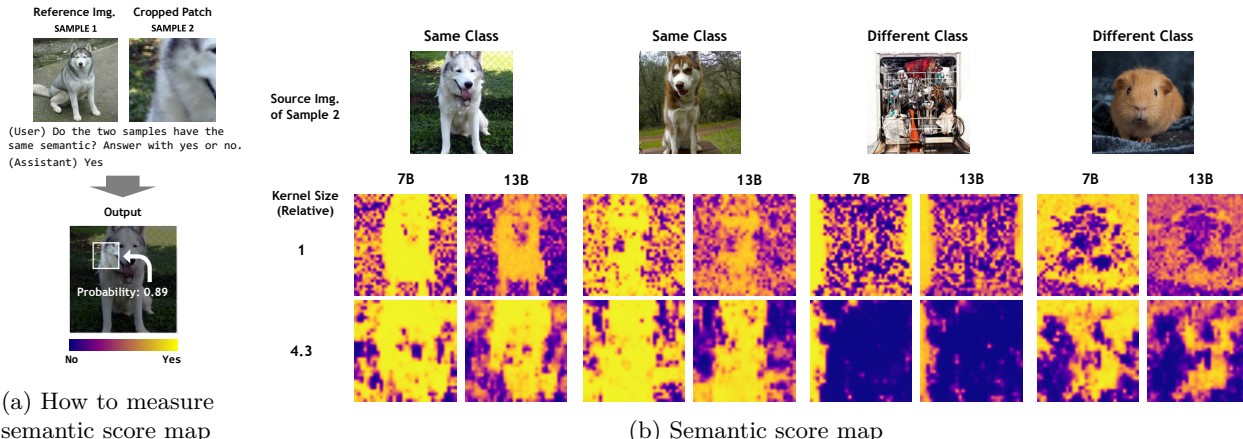

(a) How to measure semantic score map

(b) Semantic score map

Figure 7: **Patch analysis. (a)** We provide the reference image and the target patch to LLaVA, repeating the process by sliding and then extracting the probability of the "yes" token. **(b)** We fixate the reference image from (a), vary the target patch, and visualize the score map. When samples 1 and 2 belong to the same class, the scores are higher on the object itself. Notably, the smaller model tends to assign higher scores to the background areas.

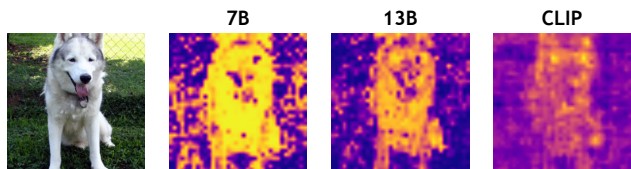

Figure 8: **Patch analysis result with visual encoder.** We compute the similarity between the reference image and patches. For the CLIP encoder, we adopt cosine similarity as a score.

the role of semantics is critical in visual perception. Thus, we investigate how well VLMs process semantic-related information. The improvement of performance on the semantics in Table 1c is noticeable in a larger model. These results raise the question of why larger models have better performance on semantics.

### 3.6 Why do VLMs have different accuracies?

Given the same visual encoder, why do the larger VLMs have higher accuracies on the semantic dataset? We hypothesize that the performance difference comes from LLM because it uses the same visual encoder. To validate this hypothesis, we visualize how LLMs make decisions. Inspired by the weakly supervised object localization method (Choe et al., 2022), we assign probability to patches (see Fig. 7a) as follows.

1. Given two images, one is used as the reference image, which is sample 1. The other is the source of a target.

2. Crop the target source image into patches according to the patch size and stride. The cropped patch is used for sample 2.

3. Ask VLMs if the reference and target patches share the same semantics and record the probability of "yes" tokens as the score.

Figure 7b shows the score map for target images according to the model size and patch size.

**Result.** Compared to the results for the same and different classes in Fig. 7b, the score maps for the same class have high scores in object-present regions, contrasting with the noisy and structure-lacking maps for different classes. The larger model assigns a lower score to the background compared to the smaller model.

Figure 8 shows the result of VLMs and the corresponding visual encoder. We observe that distinction and discrimination become sharper after passing the LLM. We think that LLMs help to better understand context and complement the visual encoder. It implies that larger LLMs achieve more accurate semantic recognition as reflected in Table 1c. Also, the kernel size affects the reliability of the captured objects and the noisiness of the score map.

Inferring whether a reference image and a patch share the same semantics requires high reasoning capabilities. Prior works (Brown et al., 2020; Zoph et al., 2022) have demonstrated that larger LLMs exhibit better reasoning abilities. Accordingly, when the vision encoder remains consistent, larger LLMs are likely to perform better in understanding and interpreting semantic information.

## 4    Potential Application

Our findings suggest that VLMs can improve image recognition by applying simple pre-processing to input images. For example, in Fig. 9, the chart reasoning results from GPT-4 (OpenAI, 2023) vary according to graph colors or symbol shapes. Regarding the color, GPT-4 struggles to distinguish between similar shades of color, making it difficult for VLMs to match colors with their corresponding numerical values. As revealed by the color sensitivity check, this result is due to the model's lack of color competency rather than its reasoning ability. By adjusting the colors, GPT-4 provides accurate responses. Regarding the shape, GPT-4 is confused between triangle and rectangle symbols, indicating a lack of shape competency that may result in incorrect recognition outcomes. When we change the rectangle to a circle, GPT-4 provides the correct reasoning output.

Our eye examination framework can also guide the selection of necessary components. For instance, tasks like anomaly detection often demand advanced perception capabilities related to shape, color, and other visual features. By designing the eye examination to the specific requirements of the task, we can effectively identify and prioritize models that contribute to demonstrating performance in examinations. We hope that our VLM's eye examination process will not only assist in selecting models and manipulating images but also facilitate enhancements in model performance by deepening our understanding of underlying factors.

## 5    Related Work

LLMs (OpenAI, 2023; Touvron et al., 2023a;b; Zhang et al., 2022b; Chowdhery et al., 2023; Roberts et al., 2020; Chiang et al., 2023) have shown a remarkable ability to process text and language information and generate rational responses. Building on the success of LLMs, VLMs (Dai et al., 2023; Liu et al., 2023b;a; Gao et al., 2023b; Zhang et al., 2024) have emerged. These models, which combine visual encoders with LLM inference capabilities, have made significant progress. Unlike LLMs, VLMs are capable of processing visual information such as images. VLM needs to interpret the information in an image based on a given task or prompt; instruction-tuned VLMs, such as LLaVA (Liu et al., 2023b) and InstructBLIP (Dai et al., 2023), have demonstrated generalizability in addressing multiple tasks. To improve our understanding of VLMs, we focus primarily on understanding how VLMs recognize and interpret visual information.

Both humans and neural networks exhibit similarities in terms of information processing (Alper et al., 2023; Zhang et al., 2022a). Numerous studies aim to understand human cognitive processes, often in highly controlled settings. For example, when developing the CIE color model (Wright-Guild experiment), participants were required to look through a small aperture, limiting their field of view to just 2 degrees. Similarly, investigations of VLMs are frequently conducted in restricted environments (Zhu & Li, 2023; Allen-Zhu & Li, 2023). Inspired by prior work, we analyze changes under limited conditions in terms of color, shape, and semantics. Our proposed LENS dataset allows us to do such an analysis.

## 6    Conclusion

We take a closer look at the fundamental perceptual abilities of VLMs through an eye examination process that focuses on color, shape, and semantics. We introduce the LENS dataset, which instructs the model

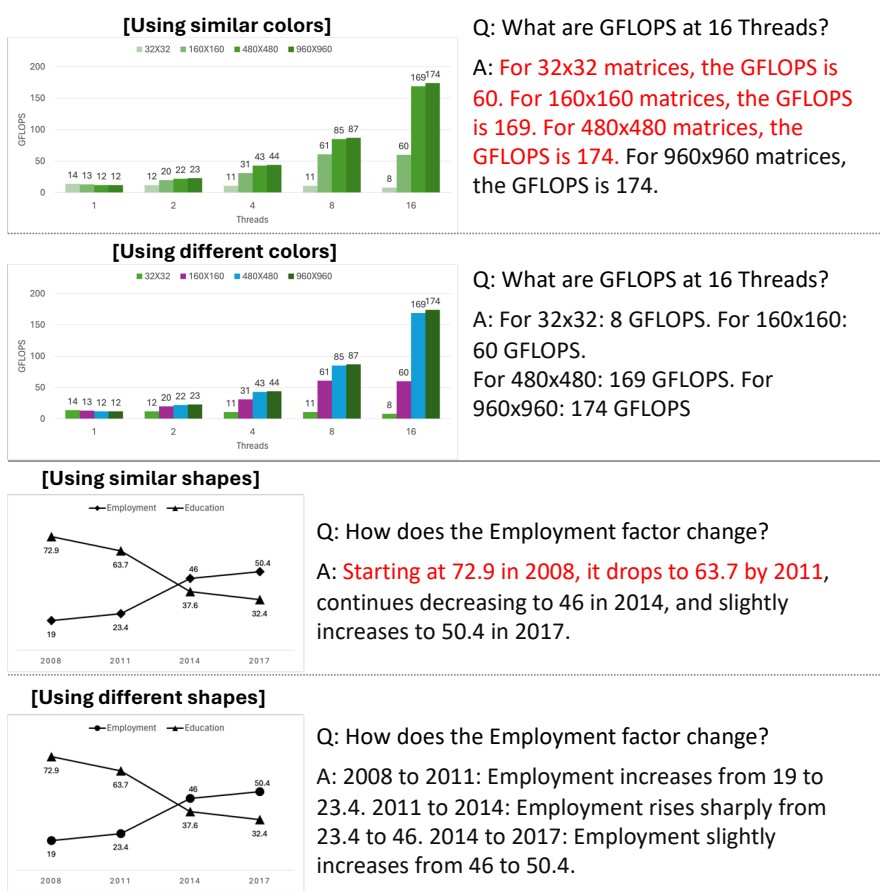

Figure 9: **Potential Application.** We test our findings in the chart reasoning from GPT-4. We vary the color of the chart or the symbol shapes. GPT-4 is a lack of understanding of the chart because of improper color and shape. After changing color and shape, we can increase the correctness.

about the examination and ensures that the model is in the appropriate state for the examination. Fine-tuning and evaluating the models on LENS allows us to conduct a detailed analysis. The color sensitivity results indicate that VLMs are less responsive to the green spectrum. We also observe that the LLM, which acts as the brain of the VLM, affects shape sensitivity and the ability to distinguish patch-wise semantics for recognition. We believe that these insights and the proposed evaluation process will contribute to a deeper understanding of VLMs' behavior and improve reasoning capabilities, as shown in the potential application.

### Acknowledgments

This work was supported by Institute of Information & communications Technology Planning & Evaluation (IITP) grant funded by the Korea government(MSIT) (No. 2022-0-00124, No.RS-2022-II220124, Development of Artificial Intelligence Technology for Self-Improving Competency-Aware Learning Capabilities; No. RS-2024-00457882, National AI Research Lab Project; No.RS-2019-II191906, Artificial Intelligence Graduate School Program(POSTECH))

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

## A   Appendix

In this appendix, we include an additional discussion, the details of our LENS and experiments, and additional qualitative results, which are not included in the main paper.

## A   Impact Statement

As research on Large Language Models (LLMs) and Vision-Language Models (VLMs) begins, there is a noticeable acceleration in the development speed within the AI industry. This advancement in research has the potential to enrich human life, yet it simultaneously brings rapid changes to our lifestyles. Despite the swift progress in large model research, a gap remains in our understanding of the precise operational principles of these systems. This lack of clarity and control over AI's deep integration into daily life is raising significant social concerns. Our project aims to address these issues, focusing specifically on analyzing the perception mechanisms of VLMs. For this reason, our goal is to move towards a more controllable system through a better understanding of VLMs by casting a fundamental question, "How do VLMs perceive the world?". However, it is important to recognize that such controllability does not always yield positive effects; it could potentially be exploited for more complex and malicious criminal activities. Currently, our analysis is in its nascent stage, but we hope that our research will contribute to the development of controllable and explainable AI, ultimately fostering a safer coexistence with AI in the world.

## B   Additional Discussion

Understanding the inside of AI is difficult because of overparameterization, non-linear mapping, non-convex optimization, etc. Many researchers have tried to provide and explain the behavior of AI in terms of theory (Allen-Zhu et al., 2019a;b), explainable algorithms (Ribeiro et al., 2016; Lundberg & Lee, 2017; Wu et al., 2023; Sundararajan et al., 2017; Petroni et al., 2019; Wu et al., 2020; Selvaraju et al., 2017), physics law (*e.g.,* scaling low) (Kaplan et al., 2020; Gao et al., 2023a; Henighan et al., 2020; Hernandez et al., 2021; Cherti et al., 2023; Bubeck & Sellke, 2021), and computational experiments (Zhang et al., 2017; Prystawski et al., 2023; Akata et al., 2023).

The approach of deep learning theory explains the phenomena of AI in terms of the mathematical form. Thus, it is more rigorous than other approaches. However, deep learning theory includes impractical assumptions such as shallow or infinite-width networks. Explainable AI is to provide an understandable form to users. For example, previous methods have revealed how and why the model's decisions are made. The approach of physics law is to find certain laws that occur in models; the popular law is the scaling law that the larger the model and the more data we use, the better the performance. Similarly, the computational approach exists by observing the model's behavior. Some methods are inspired by philosophy, psychology, and cognitive science because they are based on large language models (LLMs) showing emergent intelligence and reasoning. These approaches might not be rigorous compared to the deep learning theory. Our approach is close to the computational experiments. We fine-tuned the pre-trained large model with our dataset and investigated the behavior of the model along with the visual input changes.

## C   Details of LENS

We propose the eye examination process of VLMs to understand how the model perceives the given visual signals. The LENS (Learning ElemeNt for visual Sensory) dataset includes three types of primitive visual information: color, shape, and semantic. The statistics of our LENS are in Table 4. Each LENS data consists of an image, a question with answer options, and a ground truth answer. The questions are randomly sampled from a set associated with each category. In this section, we provide the set of questions for each category of our LENS dataset.

Additionally, in Fig. 10, we show samples for the patch group in the semantic category. For the self-swap, two randomly selected patches are swapped within an image. The goal is to find the positions of swapped patches. In contrast, for the cross-swap, a single patch is randomly sampled from each image and swapped

|  | Color | Shape | Semantic | | |
|---|---|---|---|---|---|
|  |  |  | yes or no | 1 or 2 | Patch |
| **Train** | 2,648 | 6,720 | 3,500 | 1,820 | 3,500 * 3 |
| **Validation** | 568 | 3,360 | 1,000 | 520 | 1,500 * 3 |

Table 4: **Statistics of the LENS dataset.**

between images. The goal is to find the position of a patch from a different image. For masking, we first randomly sample four patches and replace them with black patches. The goal is to find the appropriate location for one of the sampled patches.

## C.1 Color

> **The list of questions for color dataset in LENS.**
>
> **Color [Yes or No]**
> - "Are the color boxes the same color?"
> - "Do the samples have the same color?"
> - "There are two sample boxes; are the two samples have the same color?"
> - "Are the two sample boxes painted in the same color?"
> - "Provide an answer on whether the two samples are the same color."
> - "Give a short and clear answer to whether the colored boxes have the same color."
> - "Share a concise result of whether the colors of boxes are the same."
> - "Offer a succinct explanation of whether the two samples are the same color."
>
> **Color [SAMPLE 1 or SAMPLE 2]**
> - "Each sample has two colored boxes, which sample has the same color?"
> - "Each sample has two colored boxes, which sample has the boxes having the same color?"
> - "Each sample has two colored boxes, which sample has the same colored boxes?"
> - "Each sample has two colored boxes, which sample has the two boxes painted in the same color?"
> - "Each sample has two colored boxes, provide an answer on which sample is colored with the same color."
> - "Each sample has two colored boxes, give a short and clear answer to what is the sample that has the same color."
> - "Each sample has two colored boxes, share a concise result of what sample has the same color boxes."
> - "Each sample has two colored boxes, offer a succinct explanation of which sample has the same color."

## C.2   Shape

---

The list of questions for shape dataset in LENS.

**Shape [Yes or No]**
- "Are the shapes of the two given samples the same?"
- "Are the two samples identical in shape?"
- "Do both samples have the same form?"
- "Is the outline of sample 1 matching with that of sample 2?"
- "Are the shapes of these two samples alike?"
- "Is there any shape difference between the two samples?"
- "Do the samples share the same geometry?"
- "Are the forms of the two samples equivalent?"

---

The list of questions for shape dataset in LENS.

**Shape [SAMPLE 1 or SAMPLE 2]**
- "Each sample has two shapes, which sample has the same shape?"
- "Each sample has two shapes, which sample has the images having the same shape?"
- "Each sample has two shapes, what sample has the same shape?"
- "Each sample has two shapes, which sample has the two images having the same shape?"
- "Each sample has two shapes, provide an answer on which sample contains the same shape."
- "Each sample has two shapes, give a short and clear answer to what sample has the same shape."
- "Each sample has two shapes, provide a concise answer of which sample has the same shape."
- "Each sample has two shapes, offer an answer which sample has the same shape."

## C.3   Semantic

---

The list of questions for the semantic dataset in LENS.

**Semantic [Yes or No]**
- "Do the two samples have the same semantic?"
- "Do the samples have the same semantic?"
- "There are two samples; do the two samples have the same semantic?"
- "Do the two samples have the same semantic?"
- "Provide an answer on whether the two samples have the same semantic."
- "Give a short and clear answer to whether the two samples have the same semantic."
- "Provide a concise result of whether the same semantic."
- "Offer a succinct explanation of whether the two samples are the same semantic."

**Semantic [SAMPLE 1 or SAMPLE 2]**
- "Each sample has two images, which sample has the closer(or same) semantic than the other?"
- "Each sample has two images, which sample has the images having the same(or closer) semantic?"
- "Each sample has two images, which sample has the closer(or same) semantics than the other?"
- "Each sample has two images, which sample has the two images having the same(or closer) semantics?"
- "Each sample has two images, provide an answer on which sample contains the closer(or same) semantic than the other."
- "Each sample has two images, give a clear answer to which has the same(or closer) semantic."
- "Each sample has two images, provide a concise result of which sample has the closer(or same) semantic than the other."
- "Each sample has two images, offer a succinct explanation of which sample has the same(or closer) semantic."

## C.4 Patch

---

**The list of questions for patch dataset in LENS.**

**Patch Cross**
- "Within the 16 divided sections of the image, identify the single patch that originates from a different image."
- "Of the 16 segments in the given image, which one is extracted from an alternate image?"
- "There is one patch among the 16 in the image that doesn't belong; can you locate it?"
- "Spot the section among the 16 patches of the image that was taken from a distinct image."
- "A single patch out of the 16 in the provided image is from a different source. Where is it located?"
- "One patch in the 16-part divided image is mismatched from another image. Where can it be found?"
- "In the image that's split into 16 patches, which section is the outlier sourced from another image?"
- "Locate the patch that differs from the rest in the 16-part grid of the image, indicating it's from a different image."

**Patch Self**
- "Identify the two patches that have exchanged places in the 16-patch divided picture."
- "Which two patches in the picture's grid of 16 have been swapped with each other?"
- "Locate the two patches whose positions have been reversed in the image composed of 16 patches."
- "In the 16-part grid of the image, which two patches are out of their original positions?"
- "Find the pair of patches that have been interchanged in the 16-segmented image."
- "Determine the positions of the two patches that have been swapped in the 16-patch picture."
- "Spot the two patches that aren't in their correct spots within the 16 divided sections of the picture."
- "Assess the 16-patch arrangement of the image and identify the two patches that have been swapped."

**Patch Masking**
- "Given the picture divided into 16 patches with some missing, identify where the patch on the right fits."
- "In the 16-segmented picture, where does the separate patch on the right go?"
- "Where is the correct position for the patch on the right in the picture's 16-patch layout?"
- "Determine where the patch shown on the right should be placed among the missing patches in the picture."
- "Regarding the image split into 16 patches with some absent, where should the right-hand patch be located?"
- "Find the appropriate spot for the right-side patch in the grid of 16 patches where some are missing."
- "Assess the 16-patch division of the picture and place the patch from the right in its proper location."
- "Where would the patch on the right be placed in the fragmented 16-patch picture?"

---

## C.5 Fine-Tuning Setup

We finetune VLMs with our constructed dataset, color, shape, and semantics, respectively. We chose LLaVA (Liu et al., 2023b) because it is efficient in terms of training time. We employ LoRA (Hu et al., 2022) during the finetuning because it is a resource-efficient algorithm. We set the training epoch as 2, batch size 128, and learning rate 0.0002 with cosine scheduling, Adam optimizer (Kingma & Ba, 2015) and gradient checkpointing. We use 8 A100 80G GPUs for our experiments.

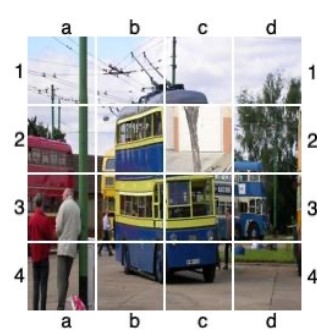

The image is divided into 16 patches, with columns
labeled from A to D and rows from 1 to 4. For example,
(1,1) is a1, (1,4) is a4, (4,1) is d1, and (4,4) is
d4. Answer the following question based on this:
{One of the questions above}
Here are options:
OPTION 1: b3, OPTION 2: c2, No Answer: none
Answer with OPTION 1 or OPTION 2 or No Answer.

(a) **Cross-Swap**

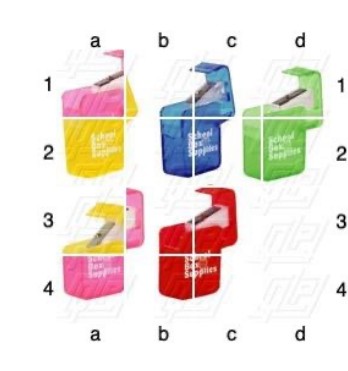

The image is divided into 16 patches, with columns
labeled from A to D and rows from 1 to 4. For example,
(1,1) is a1, (1,4) is a4, (4,1) is d1, and (4,4) is
d4. Answer the following question based on this:
{One of the questions above}
Here are options:
OPTION 1: a1 and c3, OPTION 2: a1 and a3, No Answer:
none
Answer with OPTION 1 or OPTION 2 or No Answer.

(b) **Self-Swap**

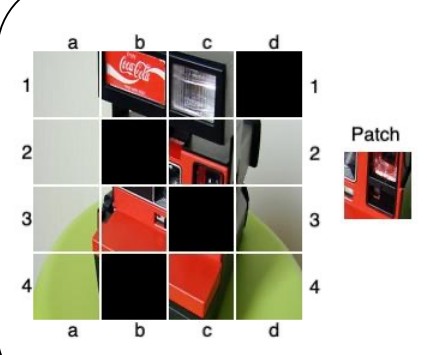

The image is divided into 16 patches, with columns
labeled from A to D and rows from 1 to 4. For example,
(1,1) is a1, (1,4) is a4, (4,1) is d1, and (4,4) is
d4. Answer the following question based on this:
{One of the questions above}
Here are options:
OPTION 1: b2, OPTION 2: c3, No Answer: none
Answer with OPTION 1 or OPTION 2 or No Answer.

(c) **Masking**

Figure 10: **QA format for patch dataset in LENS.** For every image, we randomly sampled one of the
questions listed in C.4.

# D   Color Pattern

In Sec. 3.2, we investigate the visual encoder of VLMs to understand their varying sensitivity to colors, notably the lesser sensitivity to green compared to the other colors. We evaluate CLIP ViT-L/14 336px (Radford et al., 2021) used in LLaVA-v1.5 (Liu et al., 2023a) as described in the main paper. Here, $v(\cdot)$ represents the visual encoder (CLIP ViT-L/14 336px). We extract the hidden features, excluding the class token, that are injected into LLMs and average the tokens. We measure the cosine similarity between colors as following: $sim(c_{\mathrm{ref}}, c_{\mathrm{target}}) = \frac{v(c_{\mathrm{ref}}) \cdot v(c_{\mathrm{target}})}{||v(c_{\mathrm{ref}})||_2 ||v(c_{\mathrm{target}})||_2}$. We choose the similarity measure as cosine similarity because the training method of CLIP uses pairwise cosine similarity. It is computationally infeasible to compare all colors because the number of colors is $256^3$. Thus, we divide R, G, and B into 32 bins which means the number of colors i $32^3$. Specifically, the colors are (0, 0, 0), (0, 0, 8), ..., (248, 248, 240), (248, 248, 248). Finally, we apply min-max normalization to the results of cosine similarity to ensure values fall within [0, 1], providing better visualization and interpretation.

Figure 11 shows the similarity patterns between Red, Green, Blue and the other colors; $x$-axis represents the converted color value from RGB code. The pattern of Fig. 11c is similar to the original green value of Fig. 11a. The other colored patterns look like an upside-down green shape.

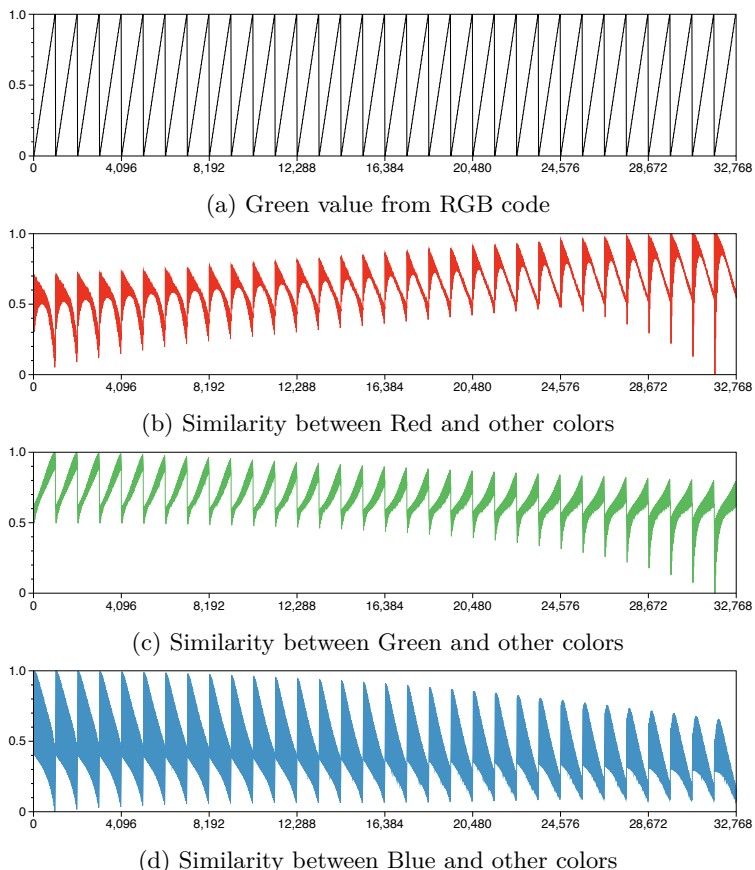

(a) Green value from RGB code

(b) Similarity between Red and other colors

(c) Similarity between Green and other colors

(d) Similarity between Blue and other colors

Figure 11: **Color similarity pattern.** We extract the color feature from CLIP ViT-L/14 336px which is used for LLaVA v1.5, and then compute the cosine similarity between the reference and target colors. **(a)**: the green value given RGB code. **(b), (c), (d)**: the cosine similarity between the {Red, Green, Blue} and target colors. We apply min-max normalization.

# E   Shape

In Fig. 12, we add the shape sensitivity graph result of InstructBLIP, Fig. 19g-19i. Since the original graph of InstructBLIP is very noisy, we perform polynomial fitting for a more comfortable comparison. As we mentioned in the main paper, InstructBLIP has a similar tendency to LLaVA, i.e., a larger model is more sensitive than a smaller one. In Table 19j, the values of SAS (sensitivity area of shape) also indicate consistent results with the graph result.

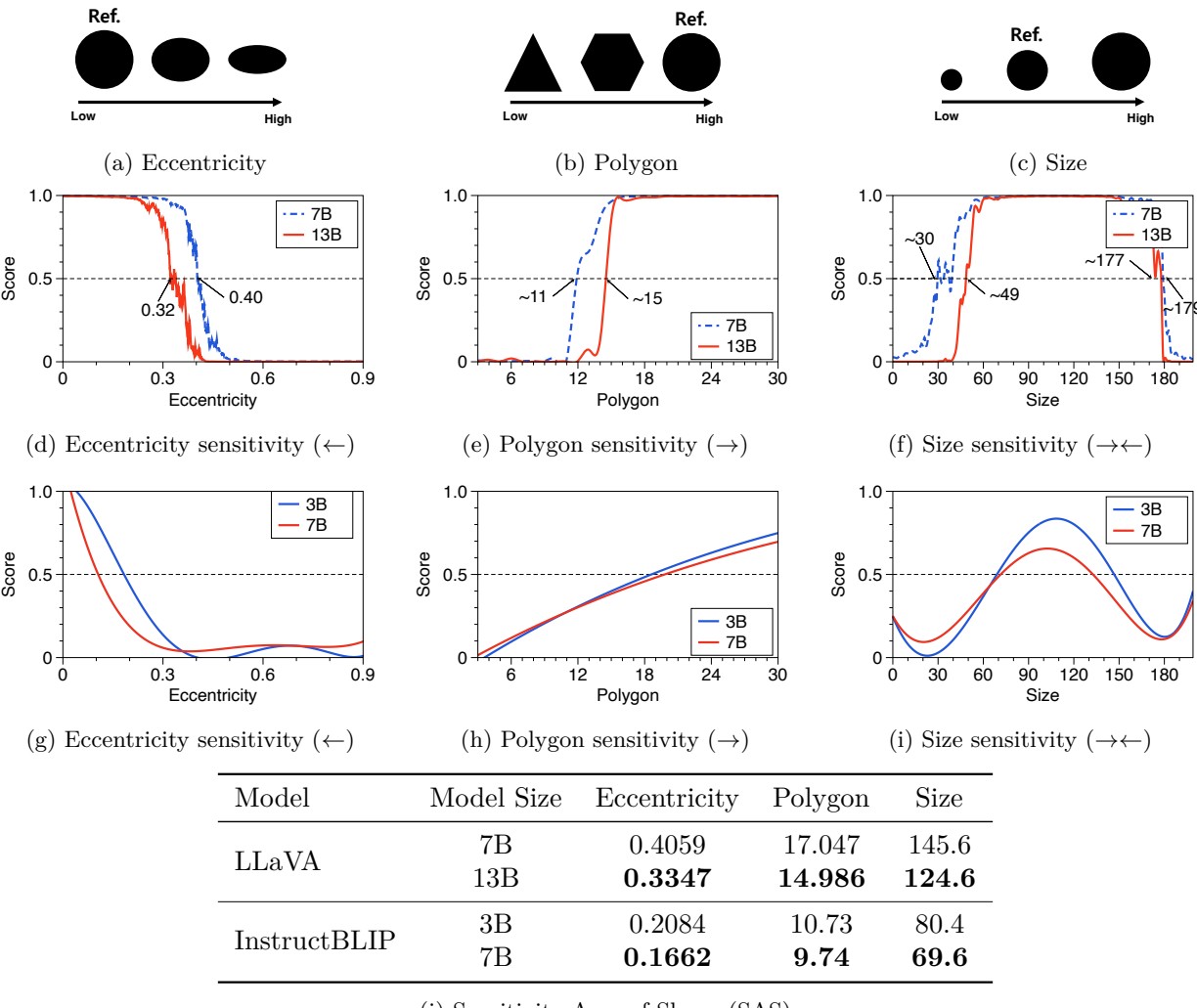

| Model | Model Size | Eccentricity | Polygon | Size |
|---|---|---|---|---|
| LLaVA | 7B | 0.4059 | 17.047 | 145.6 |
| | 13B | **0.3347** | **14.986** | **124.6** |
| InstructBLIP | 3B | 0.2084 | 10.73 | 80.4 |
| | 7B | **0.1662** | **9.74** | **69.6** |

(j) Sensitivity Area of Shape (SAS)

Figure 12: **Shape sensitivity.** We measure the sensitivity of LLaVA and InstructBLIP between a circle and target shapes by varying **(a)** the eccentricity of a circle, **(b)** the number of vertices in a regular polygon, or **(c)** the size of a circle. The graph **(d-f)** is the result of LLaVA, and **(g-i)** is of InstructBLIP. The model is more sensitive if the score changes at **(d, g)** lower eccentricity, **(e, h)** a higher number of vertices, and **(f, i)** a narrower range of size. The results shows that a larger VLM is more sensitive than a smaller one. **(j)** SAS quantifies the shape sensitivity, and shows consistent results with the graph results.

## F Patch Analysis

As mentioned in the main paper, we provide the reference image and target cropped patch to VLM, ask if the given samples are the same semantically, and visualize the score map. Figure 13 shows the results. Let the original image size be 1536; then, we vary the patch size from 260 to 60 with a fixed stride size.

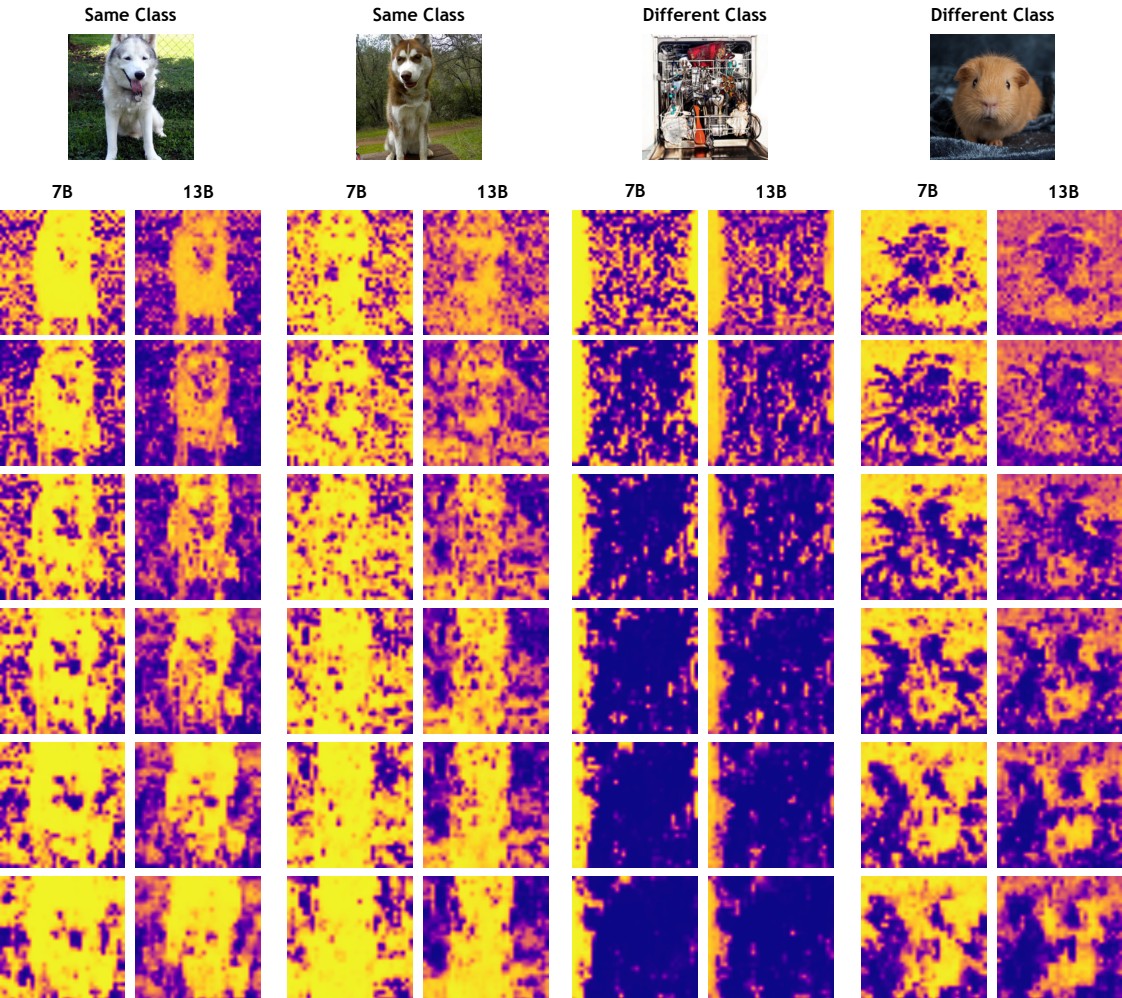

Figure 13: **Additional patch analysis result.** As mentioned in the main paper, we vary the patch size. The upper row has a smaller patch size, and the lower row has a bigger one. As the patch size increases, the score map becomes thicker because the patch can contain the object part.

## G Examination without Fine-tunning

Recent advances in VLMs demonstrate their generalization and capabilities. For example, InterVL2.5 (Chen et al., 2024a;b;c) can compare and analyze multiple images. Given the capabilities of these recent models, we provide the examination results without additional fine-tuning.

### G.1 Examination: Color

Table 5 and Figure 14 show the SAC results of InternVL2.5. Similarly, SmolVLM-Instruct shows the same tendency as shown in Table 6 and Figure 15. As shown in Table 7, Qwen2-VL-7B-Instruct demonstrates

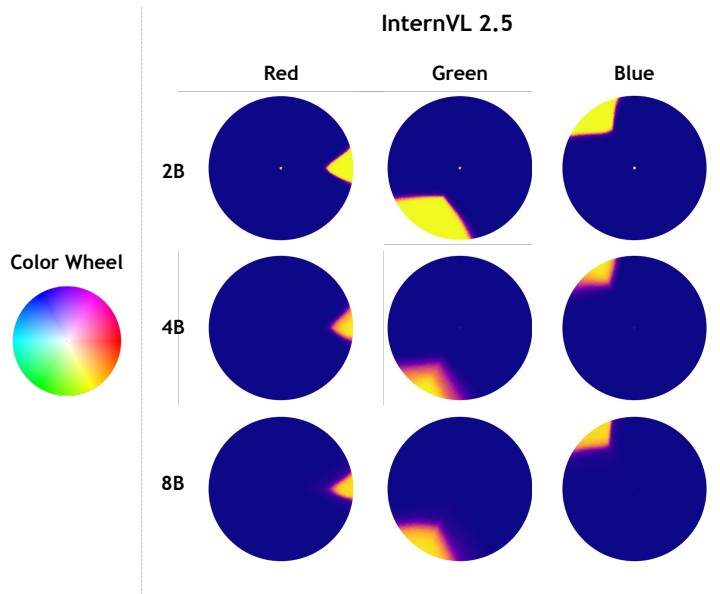

Table 5: **Sensitivity Area of Color (SAC) of InternVL2.5.**

| Model | Size | Red | Green | Blue |
|-------|------|-----|-------|------|
| InternVL2.5 | 2B | 0.1144 | 0.3953 | 0.1948 |
| | 4B | 0.0772 | 0.1827 | 0.1026 |
| | 8B | 0.0740 | 0.2023 | 0.1121 |

Figure 14: **Visualization Color sensitivity of InternVL2.5.**

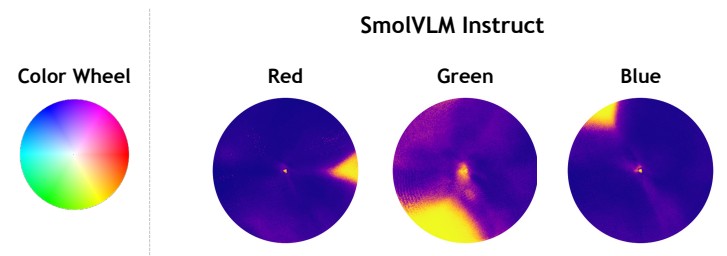

Table 6: **Sensitivity Area of Color (SAC) of SmolVLM Instruct.**

| Model | Size | Red | Green | Blue |
|-------|------|-----|-------|------|
| SmolVLM | Instruct | 0.2487 | 0.8676 | 0.2652 |

Figure 15: **Visualization Color sensitivity of SmolVLM Instruct.**

a high SAC value for green, whereas Qwen2-VL-2B-Instruct does not. To investigate this discrepancy, we visualize color sensitivity as shown in Fig. 16. We observe that, while the green area is not particularly narrow, its score value is lower than other colors. To remove the scoring intensity and compute the area, we compute the hard version of SAC as follows:

$$\text{Sensitivity Area of Color} = \int I[f(c_{\text{ref}}, c_{\text{target}}) \geq 0.5]dc_{\text{target}}, \tag{3}$$

where $I[\cdot]$ is indicate function. The values are 0.1206, 0.2760, and 0.2780 for red, green, and blue, respectively. The green and blue values are comparable.

We conduct additional experiments with the different visual encoders. Specifically, given a visual encoder, such as CLIP or SigLIP, we measure the similarity score between the target color image and the reference text "A photo of red, green, blue," instead of the reference color image. Table 8 shows the similarity score of the visual encoder. We observe that the visual encoder perceives the green color broadly.

## G.2 Examination: Shape

We visualize the score of Qwen2-VL and SmolVLM (See Fig. 18 in Appendix). The Qwen2-VL 2B model and SmolVLM Base exhibit limited comparative capabilities, as their responses yield similar scores even when the inputs are different. Figures 18 and 19 show the SAS results. In the main paper, VLMs tend to be

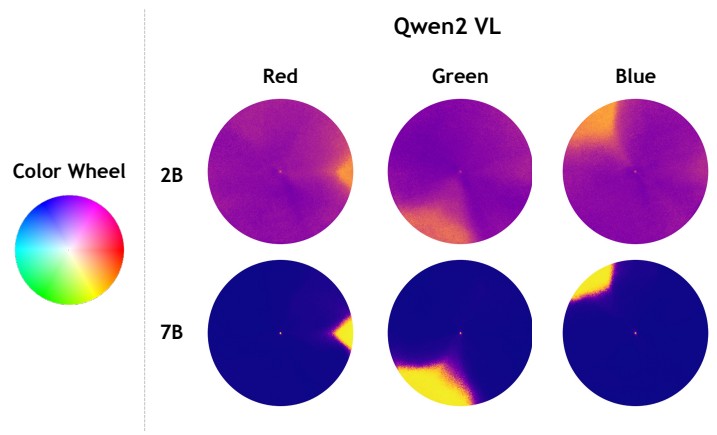

Figure 16: **Visualization Color sensitivity of Qwen2-VL.**

Table 7: **Sensitivity Area of Color (SAC) of Qwen2-VL.**

| Model | Size | Red | Green | Blue |
|---|---|---|---|---|
| Qwen2 VL | 2B | 1.0727 | 0.9949 | 1.0868 |
| | 7B | 0.1108 | 0.3920 | 0.1592 |

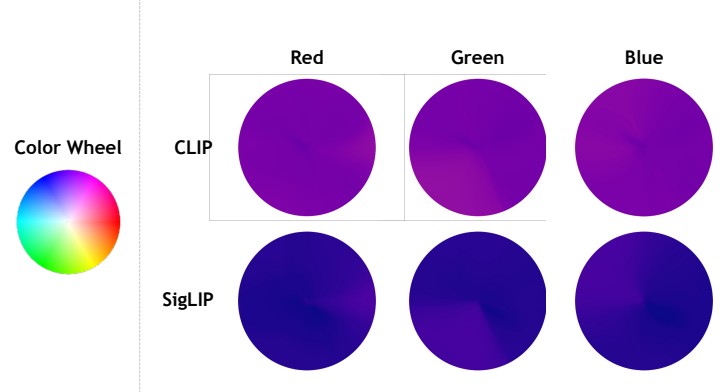

Figure 17: **Visualization color similarity score in visual encoder.** We compute the similarity score between the phrases "the photo of red, green, blue" and the corresponding target colors.

Table 8: Color similarity score in visual encoder.

| Model | Red | Green | Blue |
|---|---|---|---|
| CLIP | 0.7534 | 0.7816 | 0.7807 |
| SigLIP | 0.1333 | 0.1814 | 0.1733 |

more sensitive than smaller ones. Similarly, for InternVL2.5, the larger models generally are more sensitive compared to the smaller ones. Qwen2-VL also shows a similar overall tendency. For SmolVLM, the Instruct model shows higher sensitivity than the base model, which we attribute to its enhanced ability to follow the given instructions. The Qwen2-VL 2B model and SmolVLM Base exhibit limited comparative capabilities, as their responses yield similar scores even when the inputs are different.

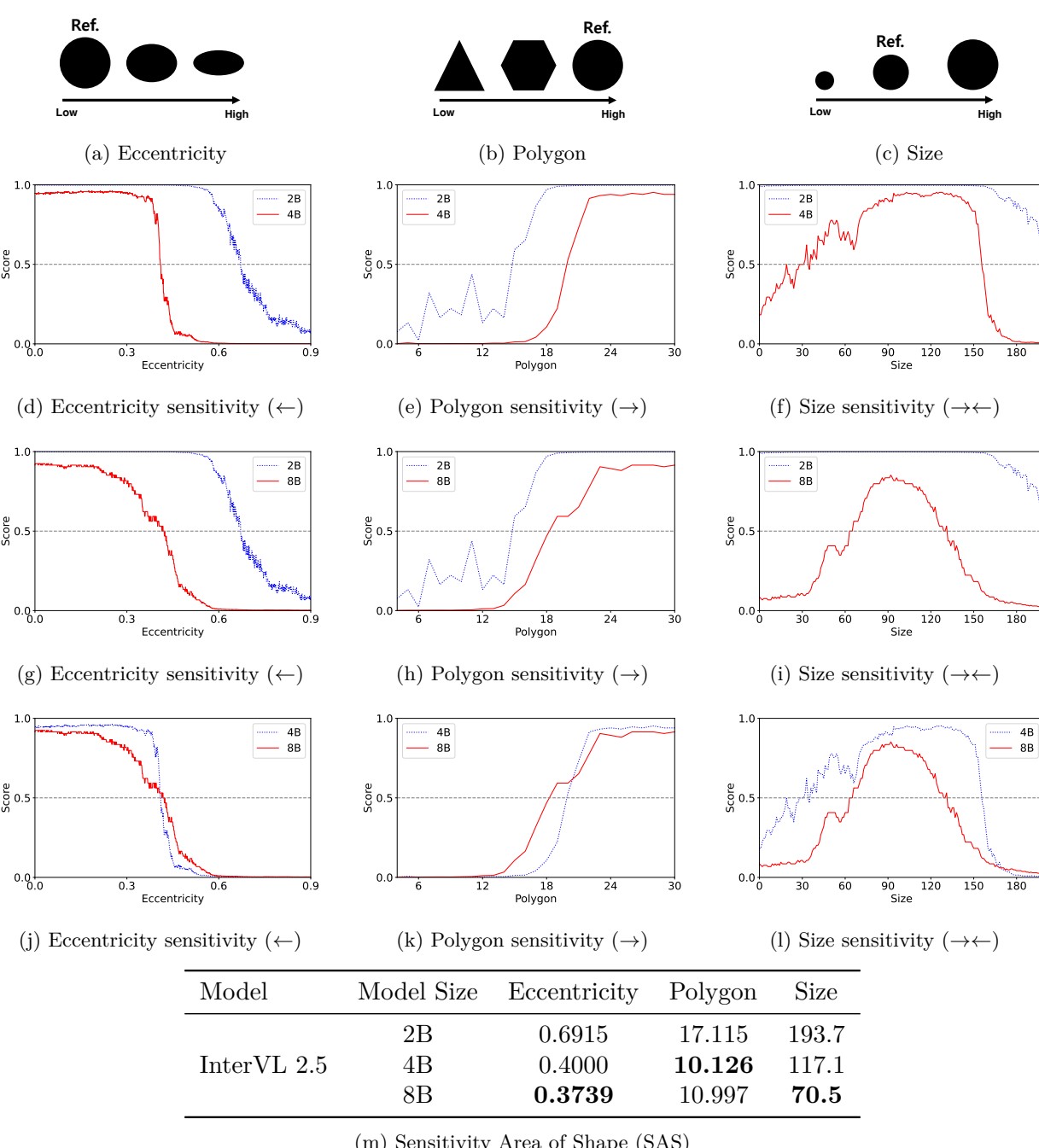

| Model | Model Size | Eccentricity | Polygon | Size |
|---|---|---|---|---|
| | 2B | 0.6915 | 17.115 | 193.7 |
| InterVL 2.5 | 4B | 0.4000 | **10.126** | 117.1 |
| | 8B | **0.3739** | 10.997 | **70.5** |

(m) Sensitivity Area of Shape (SAS)

Figure 18: **Shape sensitivity of InterVL 2.5.** The graph **(d-l)** is the result of InternVL2.5 **(m)** SAS quantifies the shape sensitivity.

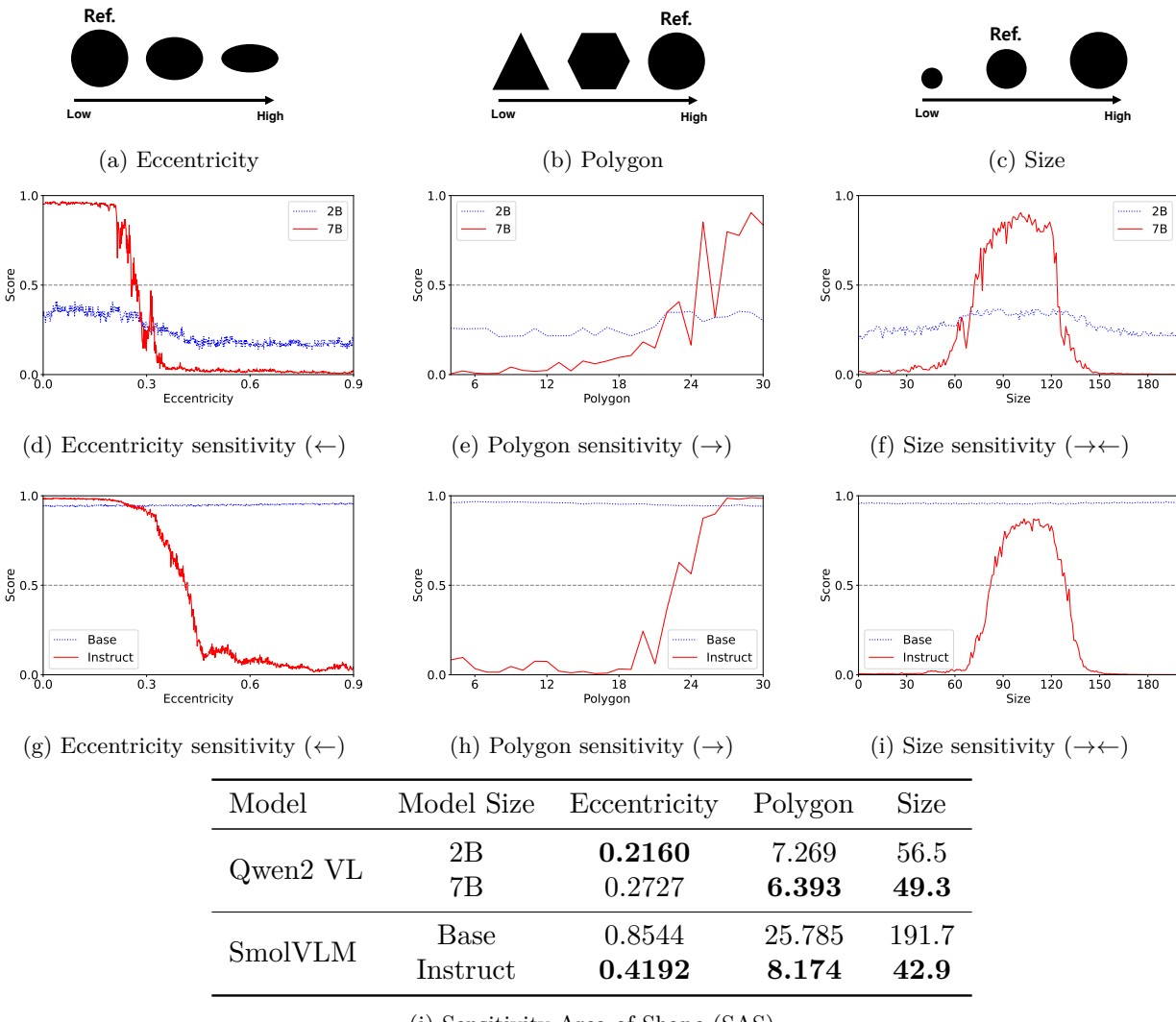

(j) Sensitivity Area of Shape (SAS)

| Model | Model Size | Eccentricity | Polygon | Size |
|---|---|---|---|---|
| Qwen2 VL | 2B | **0.2160** | 7.269 | 56.5 |
| | 7B | 0.2727 | **6.393** | **49.3** |
| SmolVLM | Base | 0.8544 | 25.785 | 191.7 |
| | Instruct | **0.4192** | **8.174** | **42.9** |

Figure 19: **Shape sensitivity of Qwen2-VL and SmolVLM.** The graph **(d-f)** is the result Qwen2-VL. The graph **(g-i)** is the result SmolVLM **(j)** SAS quantifies the shape sensitivity.

## G.3 Examination: OCR

We vary the font size within the range of 1 to 100 and assess the recognition performance based on these variations. Accuracy is computed by the exact match. Table 9 shows that the larger models are better than the smaller models. Also, we observe that the error occurs when the font size is small. If the font size is large, the models recognize it well.

## G.4 Chart

We design a small chart understanding dataset to provide the quantitative results. The colors in the chart are arranged from lowest to highest contrast. At low contrast, humans have difficulty distinguishing between labels, but as the contrast increases, the distinction becomes easier. We analyzed whether this characteristic also works in VLMs. In our experiments, the InternVL2.5 4B model starts to answer questions correctly at 80% intensity contrast, and the Qwen2 VL 7B model starts to answer correctly at 60% contrast. The results show that the higher the color contrast of the chart, the better the model understands the chart.

Table 9: **OCR ability by varying font size.** We measure the accuracy of OCR performance. We vary the font size from small to large. Models are wrong when the text is small.

| Model | Model Size | Accuracy |
|---|---|---|
| InterVL 2.5 | 2B | 97.0 |
| | 4B | 97.1 |
| | 8B | 97.3 |
| Qwen2 VL | 2B | 96.7 |
| | 7B | 97.1 |

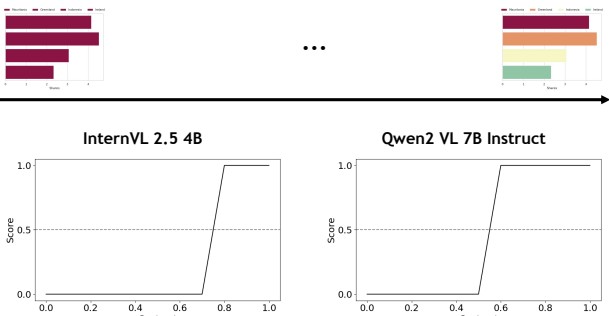

Figure 20: Chart understanding by varying contrast. We observe that the contrast is important to improve the chart understanding of VLMs.

