# OpenReview forum: "VLM’s Eye Examination: Instruct and Inspect Visual Competency of Vision Language Models"
_TMLR — Accepted by TMLR_

### Review · Reviewer_Pg4G · 2024-11-06

**Summary Of Contributions:**

The submission introduces a methodical framework dubbed the "eye examination" for assessing the visual capabilities of Vision-Language Models (VLMs). Here are the core contributions and advancements outlined in this paper:

1. **Examination Protocol**: The authors propose a structured examination protocol consisting of instruction, readiness check, and examination steps to assess the visual recognition abilities of VLMs.
2. **LENS Dataset**: The authors create the LENS (Learning Element for visual Sensory) dataset, which is designed for training and assessing VLMs' visual competencies in the areas of color, shape, and semantics.
3. **Experimental Findings**: Through extensive experiments, the authors reveal interesting insights into VLMs' sensitivities towards different visual elements, such as color, shape, and semantics.
4. **Potential Applications**: The findings of this research can inspire future improvements in VLMs and their applications in various domains, such as chart image understanding.

**Audience:**

Yes

**Claims And Evidence:**

Yes

**Requested Changes:**

The current evaluation of the two models, Llava and InstructBlip, is somewhat outdated and does not represent the latest advancements in the field. The authors are encouraged to assess more recent models such as GPT-4O, Claued35, or open-source models like Qwen2-VL/InternVL. It would be particularly interesting to see if these newer models exhibit similar patterns.

Additionally, the authors have not explored the differences among various vision encoders. Investigating whether models based on the SigLIP encoder, for instance, do not exhibit the patterns identified in this paper could provide valuable insights.

In Section 4, the discussion on potential applications raises questions about whether these applications could be extended to more powerful models. It would be beneficial to establish criteria for what types of model alterations could consistently enhance performance on images.

The exploration in Section 3.5, "Examination: Semantics," is intriguing. I am curious why larger LLMs seem to better comprehend semantic information under the same vision encoder conditions. However, I find the experiments presented insufficient; the semantic score map only indicates that larger LLMs perform well without explaining why that is the case. The authors might consider providing deeper insights, such as whether the features of large LLMs are more analogous to image features or exhibit other similarities.

**Strengths And Weaknesses:**

### Strengths
1. **Novel Evaluation Framework**: The proposed "eye examination" methodology offers a systematic and comprehensive approach to assess the visual capabilities of Vision-Language Models (VLMs). This framework draws inspiration from human vision tests and provides a structured way to evaluate VLMs' understanding of visual elements such as color, shape, and semantics.

2. **LENS Dataset**: The introduction of the LENS (Learning Element for visual Sensory) dataset is a valuable contribution. This synthetic dataset, specifically designed for instructing and assessing VLMs, categorizes visual elements into color, shape, and semantics. The dataset enables the models to understand the examination process and allows for a standardized evaluation of their visual competencies.

3. **Quantitative Metrics**: The authors define and calculate quantitative metrics, such as Sensitivity Area of Color (SAC) and Sensitivity Area of Shape (SAS), to measure VLMs' sensitivities to different visual elements. These metrics provide a clear and objective way to compare the performance of different models and understand their strengths and weaknesses.

4. **Insightful Experimental Findings**: The paper presents several interesting experimental findings that shed light on the behavior of VLMs. For instance, the authors discover that VLMs exhibit varying sensitivities to different colors, with a consistent insensitivity to green across models. They also find that larger VLMs are more sensitive to subtle shape differences compared to smaller models. These insights contribute to a deeper understanding of how VLMs perceive and process visual information.

5. **Potential Applications**: The authors discuss potential applications of their findings, such as improving VLMs' performance in chart image understanding by pre-processing the visual input based on the model's sensitivities. This demonstrates the practical relevance of the research and highlights how the proposed methodology can be used to enhance the performance of VLMs in real-world scenarios.

### Areas for Improvement
1. **Limited Model Coverage**: While the paper evaluates two popular VLMs (LLaVA and InstructBLIP), it would be beneficial to extend the analysis to a wider range of models. This would provide a more comprehensive understanding of the visual competencies across different architectures and training approaches.

2. **Robustness Analysis**: The authors could consider investigating the robustness of the proposed metrics and findings under various conditions, such as different lighting, noise levels, or image resolutions. This would help assess the generalizability of the results and identify potential limitations of the evaluation framework.

3. **Comparison with Human Vision**: Although the authors mention that VLMs' insensitivity to green differs from human vision, a more detailed comparison between VLMs and human visual perception could be informative. Exploring the similarities and differences between the two could help identify areas where VLMs need improvement to better align with human-like visual understanding.

---

> ### Author Response · Authors · 2024-12-20
> **Response to Reviewer Pg4G**
>
> We thank Reviewer Pg4G for the valuable feedback and constructive comments. Below, we provide detailed responses.
>
> **Q1. The current evaluation of the two models, LLaVA and InstructBlip, is somewhat outdated and does not represent the latest advancements in the field. The authors are encouraged to assess more recent models such as GPT-4O, Claued35, or open-source models like Qwen2-VL/InternVL. It would be particularly interesting to see if these newer models exhibit similar patterns.**
>
> We conduct the experiments using recent open-source models, specifically InternVL2.5, Qwen2-VL, and SmolVLM without fine-tuning. In summary, most existing state-of-the-art models have a similar overall trend in terms of SAC and SAS.
>
> **Q1.1 Sensitivity Area of Color (SAC)**
>
> **Table P1: Sensitivity Area of Color (SAC)**
> | Model             | Red     | Green      | Blue       |
> | ----------------- | ------- | ---------- | ---------- |
> | InternVL2.5 2B    | 0.1144  | **0.3953** | 0.1948     |
> | InternVL2.5 4B    | 0.0772  | **0.1827** | 0.1026     |
> | InternVL2.5 8B    | 0.07398 | **0.2023** | 0.1121     |
> |                   |         |            |            |
> | SmolVLM-Instruct    | 0.2487 | **0.8676** | 0.2652     |
> |                   |         |            |            |
> | Qwen2-VL-2B-Inst. | 1.0727  | 0.9949     | **1.0868** |
> | Qwen2-VL-7B-Inst. | 0.1108  | **0.3920** | 0.1592     |
> **Table P1.  Sensitivity Area of Color (SAC)**
>
> Table P1 shows the SAC results. In the main paper, green's SAC value is higher than that of other colors. The results for InternVL2.5 (2-8B) align with this trend, as the green SAC values are higher than those for other colors. Similarly, SmolVLM-Instruct shows the same tendency.
>
> Additionally, Qwen2-VL-7B-Instruct demonstrates a high SAC value for green, whereas Qwen2-VL-2B-Instruct does not. To investigate this discrepancy, we visualize color sensitivity (See Figure 16 of the appendix of the revised version). We observe that, while the green area is not particularly narrow, its score value is lower than other colors due to spread responses with lower values. To take into account more the effect of the area, we compute the hard version of SAC as follows: $\int I[f(c_{ref}, c_{target}) \ge 0.5] dc_{target}$, where $I[\cdot]$ is the indicator function. The values are 0.1206, 0.2760, and 0.2780 for red, green, and blue, respectively. The green and blue values are comparable.
>
>
> **Q1.2 Sensitivity Area of  Size (SAS)**
>
> | Model             | Eccentricity | Polygon | Size  |
> | ----------------- | ------------ | ------- | ----- |
> | InternVL2.5 2B    | 0.6915       | 17.115  | 193.7 |
> | InternVL2.5 4B    | 0.4000       | 10.126  | 117.1 |
> | InternVL2.5 8B    | 0.3739       | 10.997  | 70.5  |
> |                   |              |         |       |
> | Qwen2-VL-2B-Inst. | 0.2160       | 7.269   | 56.5  |
> | Qwen2-VL-7B-Inst. | 0.2727       | 6.393   | 49.3  |
> |                   |              |         |       |
> | SmolVLM-Base      | 0.8544       | 25.785  | 191.7 |
> | SmolVLM-Instruct  | 0.4192       | 8.174   | 42.9  |
> **Table P2: Sensitivity Area of Size (SAS)**
>
> We have added a new experiment in Figures 17-18 of the appendix of the revised version. Table P2 shows the SAS results. In the main paper, LLaVA and InstructBLIP show the trend that the larger models are more sensitive than the smaller ones. Similarly, for InternVL2.5, the larger models generally are more sensitive compared to the smaller ones. Qwen2-VL also shows a similar overall tendency. For SmolVLM, the Instruct model shows higher sensitivity than the base model, which we attribute to its enhanced ability to follow the given instructions. We visualize the score of Qwen2-VL and SmolVLM (See Fig. 19 in the appendix of the revised version). The Qwen2-VL 2B model and SmolVLM Base exhibit limited comparative capabilities, as their responses yield similar scores even when the inputs are different.

---

> ### Author Response · Authors · 2024-12-20
>
> **Q2. Additionally, the authors have not explored the differences among various vision encoders. Investigating whether models based on the SigLIP encoder, for instance, do not exhibit the patterns identified in this paper could provide valuable insights.**
>
> | Model                | Visual encoder            | Language decoder      |
> | -------------------- | ------------------------- | --------------------- |
> | InternVL2_5-1B       | InternViT-300M-448px-V2_5 | Qwen2.5-0.5B-Instruct |
> | InternVL2_5-2B       | InternViT-300M-448px-V2_5 | internlm2_5-1_8b-chat |
> | InternVL2_5-4B       | InternViT-300M-448px-V2_5 | Qwen2.5-3B-Instruct   |
> | InternVL2_5-8B       | InternViT-300M-448px-V2_5 | internlm2_5-7b-chat   |
> | SmolVLM-Base         | siglip-so400m-patch14-384 | SmolLM2-1.7B-Instruct |
> | SmolVLM-Instruct     | siglip-so400m-patch14-384 | SmolLM2-1.7B-Instruct |
> | Qwen2-VL-2B-Instruct | DFN’s ViT-675M            | Qwen-LM-1.5B          |
> | Qwen2-VL-7B-Instruct | DFN’s ViT-675M            | Qwen-LM-7.6B          |
> **Table P3. Visual encoder and language decoder of VLMs.**
>
> In response to Q1, we newly study recently proposed VLMs. In addition, Table P3 summarizes the details of the visual encoders and language decoders used in these models, highlighting their diversity.
>
> | Model      | Red    | Green  | Blue   |
> | ---------- | ------ | ------ | ------ |
> | CLIP-Large | 0.7534 | 0.7816 | 0.7807 |
> | SigLIP     | 0.1333 | 0.1814 | 0.1733 |
> **Table P4: Similarity in Visual Encoder**
>
> Furthermore, we have additionally experimented with different visual encoders. Specifically, given a visual encoder, such as CLIP or SigLip, we evaluate its sensitivity to various colors by measuring the similarity between the text “the photo of {red, green, blue}” and the corresponding target colors. Table P4 shows the sensitivity of the visual encoder. We observe that the visual encoders broadly perceive green > blue > red in order.
>
> **Q3. In Section 4, the discussion on potential applications raises questions about whether these applications could be extended to more powerful models. It would be beneficial to establish criteria for what types of model alterations could consistently enhance performance on images.**
>
> As requested, we have added additional discussion regarding criteria in Section 4.
>
> > Our eye examination framework can also guide the selection of necessary components. For instance, tasks like anomaly detection often demand advanced perception capabilities related to shape, color, and other visual features. By designing the eye examination to the specific requirements of the task, we can effectively identify and prioritize models that contribute to demonstrating performance in examinations.
>
> **Q4. The exploration in Section 3.5, "Examination: Semantics," is intriguing. I am curious why larger LLMs seem to better comprehend semantic information under the same vision encoder conditions. However, I find the experiments presented insufficient; the semantic score map only indicates that larger LLMs perform well without explaining why that is the case. The authors might consider providing deeper insights, such as whether the features of large LLMs are more analogous to image features or exhibit other similarities.**
>
> We postulate that our task on the semantic comprehension with image tokens relies on reasoning capabilities of the model: namely, inferring whether a reference image and a patch share the same semantics requires high reasoning capabilities. Prior works [C1, C2] have demonstrated that larger language models (LLMs) exhibit better reasoning abilities. Accordingly, we think that larger models have the capacity to capture and reason finer semantics in a given context when the vision encoder remains the same.
>
> [C1] Language Models are Few-Shot Learners, NeurIPS 2020
>
> [C2] Emergent Abilities of Large Language Models, TMLR 2022

---

> > ### Comment · Reviewer_Pg4G · 2024-12-21
> > **response to the author**
> >
> > It seems that even larger models or other types of vision encoders inevitably encounter the issue mentioned in the paper. Most of my concerns have been addressed, and this issue is indeed very interesting.

---

> > > ### Author Response · Authors · 2024-12-23
> > >
> > > We thank the reviewer for the valuable time in the discussion and for finding this research interesting.

---

### Review · Reviewer_6s3j · 2024-11-25

**Summary Of Contributions:**

This paper primarily investigates the visual capabilities of a vision-language model from the perspectives of color, shape, and semantics. To achieve this goal, a series of meticulously designed questions targeting these three aspects are presented. Additionally, specific metrics, such as SAC and SAS, are developed to evaluate the sensitivity to color, shape, and semantics. Following this, a detailed analysis of the results is provided. Overall, the motivation of this paper is clear, and the experiments are comprehensive.

**Audience:**

Yes

**Claims And Evidence:**

Yes

**Requested Changes:**

See weakness

**Strengths And Weaknesses:**

## Strengths

1. The motivation of this paper is clear and highly valuable. Most existing works investigate vision-language models as a whole, often overlooking the impact of visual perception abilities on overall performance.
2. The experimental designs to evaluate the sensitivity to color, shape, and semantics are novel and intriguing, and the results are comprehensive.
3. The paper is well-written and easy to read.

## Weakness

1. The procedure for fine-tuning a VLM and performing readiness checks is quite cumbersome, which may limit the adaptability of the proposed evaluation approach to other models.
2. Furthermore, the choice of evaluated VLMs, such as LLaVA and InstructBLIP, is not ideal. Most existing state-of-the-art models, such as Qwen2-VL and InternVL2, already possess prior information about color, shape, and object semantics. Additionally, they can follow the instructions in the proposed dataset.
3. Some of the formulas are quite misleading. For example, in Section 3.2, the formula
$\[ I_{x,y} = (255, 0, 0) \cdot \text{sim}(c_{\text{Red}}, I_{x,y}) + (0, 255, 0) \cdot \text{sim}(c_{\text{Green}}, I_{x,y}) + (0, 0, 255) \cdot \text{sim}(c_{\text{Blue}}, I_{x,y}) \]$
is problematic. Since $\(\text{sim}(x, y)\)$ is a scalar, it is unclear how this formula results in an image.
4. Regarding the questions in the dataset, the paper mentions that they are randomly sampled from a set. Could the authors provide detailed information about this set?
5. As the title of this paper suggests, it focuses on the visual competency of vision-language models. However, besides color, shape, and semantics, the paper overlooks an important aspect: optical character recognition (OCR) ability. The authors are encouraged to investigate this aspect as well.
6. In Section 3.1, the statement ``Considering the vast size of the color space, \(256^3 \approx 16.8M\), we reduce the color space to \(32^3 = 32,768\)`` needs clarification. What does this reduction entail?

---

> ### Author Response · Authors · 2024-12-20
> **Response to Reviewer 6s3j**
>
> We thank Reviewer 6s3j for valuable feedback and constructive comments. Below, we address each concern raised.
>
> **Q1. The procedure for fine-tuning a VLM and performing readiness checks is quite cumbersome, which may limit the adaptability of the proposed evaluation approach to other models.**
>
> These steps were necessary due to the limitations of previous models (LLaVA-1.5, InstructBLIP) in order to effectively compare two images and see if the model's capabilities have improved. However, recent advances in VLMs have improved the ability to compare images, making it possible to evaluate them directly with our eye examination without fine-tuning. We conduct experiments with recent VLMs (InternVL2.5, SmolVLM, Qwen2-VL) without fine-tuning below. Note that our dataset is still valid if models, e.g., small models, do not have capabilities for examination.
>
>
> **Q2. Furthermore, the choice of evaluated VLMs, such as LLaVA and InstructBLIP, is not ideal. Most existing state-of-the-art models, such as Qwen2-VL and InternVL2, already possess prior information about color, shape, and object semantics. Additionally, they can follow the instructions in the proposed dataset.**
>
> We conduct the experiments using recent open-source models, specifically InternVL2.5, Qwen2-VL, and SmolVLM. Recent advancements in VLMs have significantly improved their ability to compare images, allowing for a direct evaluation through our eye exams. Thus, we conduct examinations without fine-tuning as below.
>
> | Model             | Red     | Green      | Blue       |
> | ----------------- | ------- | ---------- | ---------- |
> | InternVL2.5 2B    | 0.1144  | **0.3953** | 0.1948     |
> | InternVL2.5 4B    | 0.0772  | **0.1827** | 0.1026     |
> | InternVL2.5 8B    | 0.07398 | **0.2023** | 0.1121     |
> |                   |         |            |            |
> | SmolVLM-Instruct    | 0.2487 | **0.8676** | 0.2652     |
> |                   |         |            |            |
> | Qwen2-VL-2B-Inst. | 1.0727  | 0.9949     | **1.0868** |
> | Qwen2-VL-7B-Inst. | 0.1108  | **0.3920** | 0.1592     |
> **Table P1.  Sensitivity Area of Color (SAC)**
>
> We have added a new experiment in Tables 5-7 of the appendix of the revised version. Table P1 shows the SAC results. In the initial submission, the sensitivity of green is higher than that of other colors. The results for InternVL2.5 (2-8B) align with this trend, as the green SAC values are higher than those for other colors. Similarly, SmolVLM-Instruct shows the same tendency.
>
> Additionally, Qwen2-VL-7B-Instruct demonstrates a high SAC value for green, whereas Qwen2-VL-2B-Instruct does not. To investigate this discrepancy, we visualize color sensitivity (See Figure 16 of the appendix of the revised version). We observe that, while the green area is not particularly narrow, its score value is lower than other colors due to spreaded responses with lower values. To take into account more the effect of the area, we compute the hard version of SAC as follows: $\int I[f(c_{ref}, c_{target}) \ge 0.5] dc_{target}$, where $I[\cdot]$ is the indicator function. The values are 0.1206, 0.2760, and 0.2780 for red, green, and blue, respectively. The green and blue values are comparable.
>
>
> | Model             | Eccentricity | Polygon | Size  |
> | ----------------- | ------------ | ------- | ----- |
> | InternVL2.5 2B    | 0.6915       | 17.115  | 193.7 |
> | InternVL2.5 4B    | 0.4000       | 10.126  | 117.1 |
> | InternVL2.5 8B    | 0.3739       | 10.997  | 70.5  |
> |                   |              |         |       |
> | SmolVLM-Base      | 0.8544       | 25.785  | 191.7 |
> | SmolVLM-Instruct  | 0.4192       | 8.174   | 42.9  |
> |                   |              |         |       |
> | Qwen2-VL-2B-Inst. | 0.2160       | 7.269   | 56.5  |
> | Qwen2-VL-7B-Inst. | 0.2727       | 6.393   | 49.3  |
> **Table P2: Sensitivity Area of Size (SAS)**
>
> We have added a new experiment in Figures 17-18 of the appendix of the revised version. Table P2 shows the SAS results. In the main paper, LLaVA and InstructBLIP show the trend that the larger models are more sensitive than the smaller ones. Similarly, for InternVL2.5, the larger models generally are more sensitive compared to the smaller ones. Qwen2-VL also shows a similar overall tendency. For SmolVLM, the Instruct model shows higher sensitivity than the base model, which we attribute to its enhanced ability to follow the given instructions. We visualize the score of Qwen2-VL and SmolVLM (See Fig. 19 in the appendix of the revised version). The Qwen2-VL 2B model and SmolVLM Base exhibit limited comparative capabilities, as their responses yield similar scores even when the inputs are different.
>
> In summary,  most existing state-of-the-art models have a similar overall trend in terms of SAC and SAS.

---

> ### Author Response · Authors · 2024-12-20
>
> **Q3. Some of the formulas are quite misleading. For example, in Section 3.2, the formula is problematic. Since $sim(x, y)$ is a scalar, it is unclear how this formula results in an image.**
>
> Although  $sim(x, y)$ is a scalar, the final output is a vector because the scalar value is multiplied by vectors when computing $I_{x,y}$ in the next paragraph. Thus, we can transform the original RGB value to the color-corrected value. We thank the reviewer for the comment.
>
> **Q4. Regarding the questions in the dataset, the paper mentions that they are randomly sampled from a set. Could the authors provide detailed information about this set?**
>
> We provide the detailed information below.
> For example, when requesting a color comparison, the phrasing of the question can vary while maintaining the same meaning. To enhance the diversity of questions, we define a set of questions in Section C of the appendix of the initial submission and then randomly sample from these pre-defined questions.
>
> **Q5. As the title of this paper suggests, it focuses on the visual competency of vision-language models. However, besides color, shape, and semantics, the paper overlooks an important aspect: optical character recognition (OCR) ability. The authors are encouraged to investigate this aspect as well.**
>
>
> | Model             | Accuracy |
> | ----------------- | -------- |
> | InternVL2.5 2B    | 97.0     |
> | InternVL2.5 4B    | 97.1     |
> | InternVL2.5 8B    | 97.3     |
> |                   |          |
> | Qwen2-VL-2B-Inst. | 96.7     |
> | Qwen2-VL-7B-Inst. | 97.1     |
>
> **Table P3: OCR ability. We vary font sizes from 1 to 100 and compute accuracy by matching the exact size.**
>
> We vary the font size within the range of 1 to 100 and assess the recognition performance based on these variations. Accuracy is computed by the exact match. Table P3 shows that the larger models are better than the smaller models. Also, we observe that the error occurs when the font size is small. If the font size is large, the models recognize it well. We have added the experiment in Table 9 of the appendix of the revised version.
>
> **Q6. In Section 3.1, the statement Considering the vast size of the color space, \(256^3 \approx 16.8M\), we reduce the color space to \(32^3 = 32,768\) needs clarification. What does this reduction entail?**
>
> We exhaustively compute all the responses for possible combinations of colors in 24-bit color space, which is computationally intensive. Thus, we reduce the color space for computational efficiency. We approximate the original color space by quantization.
> The RGB color model is typically represented by the vector [r, g, b], where each component (r, g, b) can take values from 0 to 255. This allows for a total of $256^3$ possible colors. However, instead of utilizing the full range, we limit the values of r, g, and b to the compact discrete set [0, 8, 16, ..., 248]. As a result, the total number of possible colors is reduced to $32^3$.
>
> ------
>
> We appreciate the reviewer's valuable feedback. We have conducted additional experiments, which are now included in the appendix of the main paper. We will revise and reorganize the relevant sections to integrate these new experiments.

---

### Review · Reviewer_A6mV · 2024-12-09

**Summary Of Contributions:**

Vision language models (VLMs) have shown impressive performance on a wide range of benchmarks. However, it remains unclear about the visual perception abilities of these VLMs. This work introduces LENS dataset to measure the perception sensitivities of VLMs on colors, shapes, and semantics. Based on the results, LLaVA and InstructBLIP are less responsive to green, which is different from human perception (i.e., humans are more sensitive to green, among red/green/blue). Also, the results demonstrate that the LLM backbones are important components for shape and semantic sensitivity. Last, the work shows an application potential on chart understanding based on the observed patterns.

**Audience:**

Yes

**Claims And Evidence:**

No

**Requested Changes:**

1. [Minor] Missing citation [1] in section 3.2.

[1] Patel, Roma, and Ellie Pavlick. "Mapping language models to grounded conceptual spaces." International conference on learning representations. 2022.

**Strengths And Weaknesses:**

**Strengths**
1. The idea of measuring sensitivity towards color, shape, and semantics is interesting.
2. The results of sensitivity of VLMs towards green are interesting, since this is different from human’s perception pattern.
3. The application on chart understanding is straightforward and looks promising.
4. The paper is well-written and easy to understand.


**Weaknesses**
1. Table 1 lacks a human baseline to compare with in order to understand the visual perception abilities of VLMs compared with humans. The human baseline can be more interesting for color sensitivity because it shows a different pattern between human perception and VLM’s perception. Also, Table 1 should highlight the rows which stand for “no finetuning” and “with finetuning”.
2. Figure 3 misses a human baseline. This can demonstrate how humans perceive colors in a straightforward way.
3. In Section 3.2, do the reference images contain only the red or green or blue pixels?
4. For color sensitivity analysis (Section 3.1 and Figure 3), the analysis mainly focuses on the whole VLMs. The work can also study the visual perception of the visual encoders by measuring cosine similarity scores between reference color and target color from the visual encoder’s outputs.
5. Similar to the above. For color sensitivity analysis (Section 3.2), when computing the similarity scores, the v(.) score can be extended to the last layer hidden representations of image patches from the LLM (you can ignore the text sequence and pass in only the image patches to the LLM for this). In this way, the work can compare and see the impact of LLMs during color perception.
6. In Table 1, it seems that larger InstructBlip cannot outperform smaller ones across multiple tasks. Why does this happen?
7. I understand that finetuning can let the VLMs know how to compare two sample images better. However, this goes against with the idea of understanding the perception abilities of VLMs, since the model’s perception abilities may improve during the finetuning, and thus the current results (based on finetuned VLMs on LENS) cannot demonstrate the visual perception abilities of the off-the-shelf VLMs. Can we run the experiments with off-the-shelf VLMs?
8. For Figure 2, the work can add the analyses based on just the visual encoders to reflect how LLM impacts the visual processing. The work can compute cosine similarity scores between the source image and the target patch to make the plots. Plus, it will be meaningful to add the predicted class labels of each source image to see when the model assigns higher scores to the background, will the model fail to recognize the object in the image.
10. Is it possible to adopt the findings to a small chart understanding benchmark to better demonstrate the potential of application on chart understanding? This can quantitatively demonstrate the potential of application of the findings.

---

> ### Author Response · Authors · 2024-12-20
> **Response to Reviewer A6mV**
>
> We thank Reviewer A6mV for the valuable feedback and constructive comments. Below, we address each comment.
>
> **Q1-1. Table 1 lacks a human baseline to compare with in order to understand the visual perception abilities of VLMs compared with humans. The human baseline can be more interesting for color sensitivity because it shows a different pattern between human perception and VLM’s perception.**
>
> We thank Reviewer A6mV for the valuable suggestion. We discussed human color perception in the Results of Section 3.1. We have also included additional references [C1, C2] in cognitive sciences in the caption for Figure 3.
>
>
> Prior works [C1, C2] have investigated the color sensitivity of the human eye, demonstrating that peak sensitivity occurs at wavelengths of 550–560 nm. The human eye is most responsive to yellow-green colors due to the characteristics of cone cells, which is the opposite result of the VLMs.
>
> While we agree that adding a human baseline is interesting, conducting such human subject experiments requires another completely different and redundant research with [C1,C2] requiring a highly sophisticated setup and a controlled environment, which becomes cognitive science research not capable given the limited revision period.  Instead, we have reflected the reviewer’s comment by discussing the relevant literature.
>
> Again, we thank Reviewer A6mV for the valuable suggestion.
>
> [C1] Fluorescent Penetrant Sensitivity and Removability - What the Eye Can See, a Fluorometer Can Measure, Materials Evaluation 1984
>
> [C2] Faughn, Jerry S. & Raymond A. Serway. College Physics, 6th ed. Canada: Thomson, Brooks/Cole, 2003: 675.
>
> **Q1-2. Table 1 should highlight the rows which stand for “no finetuning” and “with finetuning”**
>
> Regarding the suggestion about the table, we have highlighted the rows of Table 1. We thank Reviewer A6mV for the comment to improve the clarity of the table.
>
> **Q2. Figure 3 misses a human baseline. This can demonstrate how humans perceive colors in a straightforward way.**
>
> While we acknowledge the value of a human baseline, the same as the response to Q1-1, we believe that experimental setups should be carefully designed in controlled environments and be redundant with the long-standing and well-known cognitive science findings. For the detail, please refer to the response to Q1-1.
>
>
> **Q3. In Section 3.2, do the reference images contain only the red or green or blue pixels?**
>
> Yes. Reference images contain only the red, green, or blue pixels in Section 3.2.
>
> **Q4. For color sensitivity analysis (Section 3.1 and Figure 3), the analysis mainly focuses on the whole VLMs. The work can also study the visual perception of the visual encoders by measuring cosine similarity scores between reference color and target color from the visual encoder’s outputs.**
>
>
> | Model      | Red    | Green  | Blue   |
> | ---------- | ------ | ------ | ------ |
> | CLIP-Large | 0.7534 | 0.7816 | 0.7807 |
> | SigLIP     | 0.1333 | 0.1814 | 0.1733 |
> **Table P1: Similarity in Visual Encoder**
>
> We have added a requested experiment in Table 8 of the revised version (Table P1) that studies the color sensitivity in the visual encoder. To study how broadly the visual encoder can perceive the red, green, and blue colors, we measure the similarity score between the target color image and the reference text “A photo of {red, green, blue},” instead of the reference color image.
> Table P1 shows that the similarity of green is higher than other colors. In other words, visual encoders are less sensitive to green than red or blue, which is consistent with the results of the whole VLM in Table 2 of the initial submission.
>
> **Q5. Similar to the above. For color sensitivity analysis (Section 3.2), when computing the similarity scores, the v(.) score can be extended to the last layer hidden representations of image patches from the LLM (you can ignore the text sequence and pass in only the image patches to the LLM for this). In this way, the work can compare and see the impact of LLMs during color perception.**
>
> To respond to the comment, given that a single image provides multiple patches, we refer to methods used for extracting embeddings from LLMs [C3, C4]. Specifically, we use the following prompt: "In one word, describe the image:" and extract the hidden representation from the last layer. This approach aligns well with the method the reviewer described in the comment, with the addition of the prompt inspired by prior works [C3, C4].
>
> ~~We are currently running the experiment hard, and during the rebuttal period, we will provide a response when the experiment is completed; otherwise, we promise to include it in the final version.~~ Please see the corresponding result in the comment below.
>
> [C3] Your Mixture-of-Experts LLM Is Secretly an Embedding Model For Free, 2024
>
> [C4] Scaling Sentence Embeddings with Large Language Models, EMNLP, 2024

---

> ### Author Response · Authors · 2024-12-20
>
> **Q6. In Table 1, it seems that larger InstructBlip cannot outperform smaller ones across multiple tasks. Why does this happen?**
>
> The same trends are also observed in Table 1 of the original InstructBLIP paper [C5], where the performance of the smaller model (InstructBLIP with vicuna-7B) is better than the larger one (InstructBLIP with vicuna-13B) on several benchmarks. For example, the accuracies of the smaller and larger models are 52.2 and 51.0 on the iVQA dataset, respectively. However, after fine-tuning, the larger model outperforms the smaller one due to the larger capacity to learn additional knowledge.
>
> [C5] Dai et al., "InstructBLIP: Towards General-purpose Vision-Language Models with Instruction Tuning," NeurIPS, 2023
>
>
> **Q7. I understand that finetuning can let the VLMs know how to compare two sample images better. However, this goes against with the idea of understanding the perception abilities of VLMs, since the model’s perception abilities may improve during the finetuning, and thus the current results (based on finetuned VLMs on LENS) cannot demonstrate the visual perception abilities of the off-the-shelf VLMs. Can we run the experiments with off-the-shelf VLMs?**
>
> As requested, we have newly conducted experiments using recent off-the-shelf VLMs, specifically InternVL2.5, Qwen2-VL, and SmolVLM. Recent advancements in VLMs have improved their ability to compare images without fine-tuning (which was not supported at the time of our initial submission period), allowing for a direct evaluation through our eye exams. Thus, we present new examinations without fine-tuning as below.
>
> | Model             | Red     | Green      | Blue       |
> | ----------------- | ------- | ---------- | ---------- |
> | InternVL2.5 2B    | 0.1144  | **0.3953** | 0.1948     |
> | InternVL2.5 4B    | 0.0772  | **0.1827** | 0.1026     |
> | InternVL2.5 8B    | 0.07398 | **0.2023** | 0.1121     |
> |                   |         |            |            |
> | SmolVLM-Instruct    | 0.2487 | **0.8676** | 0.2652     |
> |                   |         |            |            |
> | Qwen2-VL-2B-Inst. | 1.0727  | 0.9949     | **1.0868** |
> | Qwen2-VL-7B-Inst. | 0.1108  | **0.3920** | 0.1592     |
> **Table P2.  Sensitivity Area of Color (SAC)**
>
> Table P2 shows the SAC results of off-the-shelf VLMs. We have added the experiments of each model in Tables 5-7 of the appendix of the revised version. In the initial submission, the sensitivity of green was higher than that of other colors. The results for InternVL2.5 (2-8B) align with this trend, as the green SAC values are higher than those for other colors. Similarly, SmolVLM-Instruct shows the same tendency.
>
> Additionally, Qwen2-VL-7B-Instruct demonstrates a high SAC value for green, whereas Qwen2-VL-2B-Instruct does not. To investigate this discrepancy, we visualize color sensitivity (See Figure 16 of the appendix of the revised version). We observe that, while the green area is not particularly narrow, its score value is lower than other colors. To remove the scoring intensity and compute the area, we compute the hard version of SAC as follows: $\int I[f(c_{ref}, c_{target}) \ge 0.5] dc_{target}$, where $I[\cdot]$ is the indicator function. The values are 0.1206, 0.2760, and 0.2780 for red, green, and blue, respectively. The green and blue values are comparable.
>
>
> | Model             | Eccentricity | Polygon | Size  |
> | ----------------- | ------------ | ------- | ----- |
> | InternVL2.5 2B    | 0.6915       | 17.115  | 193.7 |
> | InternVL2.5 4B    | 0.4000       | 10.126  | 117.1 |
> | InternVL2.5 8B    | 0.3739       | 10.997  | 70.5  |
> |                   |              |         |       |
> | SmolVLM-Base      | 0.8544       | 25.785  | 191.7 |
> | SmolVLM-Instruct  | 0.4192       | 8.174   | 42.9  |
> |                   |              |         |       |
> | Qwen2-VL-2B-Inst. | 0.2160       | 7.269   | 56.5  |
> | Qwen2-VL-7B-Inst. | 0.2727       | 6.393   | 49.3  |
> **Table P3: Sensitivity Area of Size (SAS)**
>
> Table P3 shows the SAS results. We have added these experimental results in Figures 18-19 of the appendix of the revised version. In the main paper, LLaVA and InstructBLIP show the trend that the larger models are more sensitive than the smaller ones. Similarly, for InternVL2.5, the larger models generally are more sensitive compared to the smaller ones. Qwen2-VL also shows a similar overall tendency. For SmolVLM, the Instruct model shows higher sensitivity than the base model, which we attribute to its enhanced ability to follow the given instructions. We visualize the score of Qwen2-VL and SmolVLM (See Fig. 19 in the appendix of the revised version). The Qwen2-VL 2B model and SmolVLM Base exhibit limited comparative capabilities, as their responses yield similar scores even when the inputs are different.
>
> In summary, off-the-shelf VLMs have a similar overall trend in terms of SAC and SAS.

---

> ### Author Response · Authors · 2024-12-20
>
> **Q8 and Q9.**
>
> ~~During the rebuttal period, we will provide a response if the experiment is completed; otherwise, we will include it in the final version.~~ Please see the responses in the comments below. We sincerely appreciate the valuable comments.
>
> **Q10. [Minor] Missing citation [1] in section 3.2.**
>
> We thank the reviewer for providing us with related work [1]. We have cited the paper in Section 3.2 of the paper.

---

> ### Author Response · Authors · 2024-12-23
>
> **Q5. Similar to the above. For color sensitivity analysis (Section 3.2), when computing the similarity scores, the v(.) score can be extended to the last layer hidden representations of image patches from the LLM (you can ignore the text sequence and pass in only the image patches to the LLM for this). In this way, the work can compare and see the impact of LLMs during color perception.**
>
> Considering that a single image provides multiple patches, we adopt a method similar to those used for extracting embeddings from LLMs [C3, C4]. Specifically, we use the following prompt: "In one word, describe the image:" and extract the hidden representation from the last layer. This approach aligns with the method described in the comment, with the addition of the prompt inspired by prior works [C3, C4].
>
> [C3] Your Mixture-of-Experts LLM Is Secretly an Embedding Model For Free, 2024
>
> [C4] Scaling Sentence Embeddings with Large Language Models, EMNLP, 2024
>
>
>
> | Model             | Red     | Green      | Blue       |
> | ----------------- | ------- | ---------- | ---------- |
> | InternVL2.5 2B    | 2.77  | 2.59 | 2.63     |
> | InternVL2.5 4B    | 2.97  | 2.99 | 3.01     |
> | InternVL2.5 8B    | 2.97 | 3.00 | 3.02     |
> |                   |         |            |            |
> | Qwen2-VL-2B-Inst. | 3.18  | 3.17     | 3.18 |
> | Qwen2-VL-7B-Inst. | 3.18  | 3.17 | 3.17     |
> **Table P3: Similarity score of the last layer hidden representation**
>
> All colors have similar values, as shown in Table P3. The representation of the last layer is close to discrete representations because the last layer is the representation just before it is transformed into language. These discrete representations may limited to represent all colors because people have not assigned names to all the existing colors. We think that our proposed explanation of showing two samples and comparing two samples is suitable for comparing continuous colors.
>
> **Q8. For Figure 2, the work can add the analyses based on just the visual encoders to reflect how LLM impacts the visual processing. The work can compute cosine similarity scores between the source image and the target patch to make the plots. Plus, it will be meaningful to add the predicted class labels of each source image to see when the model assigns higher scores to the background, will the model fail to recognize the object in the image.**
>
> We have added a new experiment in Figure 8 of the revised version. We compute the similarity between the reference image and patches. For the visual encoder, we adopt cosine similarity as a score. We observe that distinction and discrimination become sharper after passing the LLM. We think that LLMs help to better understand context and complement the visual encoder.
>
> We thank Reviewer A6mV for the insightful suggestion for the analysis of the visual encoder.
>
> **Q9. Is it possible to adopt the findings to a small chart understanding benchmark to better demonstrate the potential of application on chart understanding? This can quantitatively demonstrate the potential of application of the findings.**
>
> Given the limited time, we have constructed a small chart understanding dataset for evaluation to run the quantitative experiment. The colors in the chart are arranged from lowest to highest contrast. At low contrast, humans have difficulty distinguishing between labels, but as the contrast increases, the distinction becomes easier. We analyze whether this characteristic also works in VLMs.
> In our experiments, the InternVL2.5 4B model starts to answer questions correctly at 80% intensity contrast, and the Qwen2 VL 7B model starts to answer correctly at 60% contrast. The results show that the higher the color contrast of the chart, the better the model understands the chart.
>
> We have added the new experiment in Figure 20 of the appendix of the revised paper. Thanks for the suggestion.
>
> ----
>
> We appreciate the reviewer's valuable feedback. We have conducted additional experiments, which are now included in the appendix of the main paper. We will revise and reorganize the relevant sections to integrate these new experiments.

---

> > ### Comment · Reviewer_A6mV · 2024-12-23
> >
> > Thank you for the rebuttal. This addresses majority of my concerns, and I believe these additional experimental results can make the work more complete and interesting. I will raise my rating.

---

> ### Author Response · Authors · 2024-12-24
>
> We thank the reviewer's valuable time and insightful comments on our work. We believe that the new experimental results suggested by the reviewer further clarify and improve our work.

---

### Decision · Action_Editor_hKRi · 2025-02-26

**Recommendation:** Accept as is

**Comment:**

Reviewers were unanimous that this paper should be accepted, with the majority of their concerns resolved during the discussion period.

**Audience:**

Novel VLM evaluations have a wide audience of interest, particularly when written in such a clear and straightforward manner.

**Claims And Evidence:**

There's consensus that this work provides clear evidence for its claims (the only pushback has been on things like missing human baselines, but those concerns were resolved).